_Article_

# Efficient search, mapping, and optimization of multi-protein genetic systems in diverse bacteria

Iman Farasat[1], Manish Kushwaha[2], Jason Collens[2], Michael Easterbrook[1], Matthew Guido[1] & Howard M Salis[1,2,*]

## Abstract

Developing predictive models of multi-protein genetic systems to understand and optimize their behavior remains a combinatorial challenge, particularly when measurement throughput is limited. We developed a computational approach to build predictive models and identify optimal sequences and expression levels, while circumventing combinatorial explosion. Maximally informative genetic system variants were first designed by the RBS Library Calculator, an algorithm to design sequences for efficiently searching a multi-protein expression space across a > 10,000-fold range with tailored search parameters and well-predicted translation rates. We validated the algorithm's predictions by characterizing 646 genetic system variants, encoded in plasmids and genomes, expressed in six gram-positive and gram-negative bacterial hosts. We then combined the search algorithm with system-level kinetic modeling, requiring the construction and characterization of 73 variants to build a sequence-expression-activity map (SEAMAP) for a biosynthesis pathway. Using model predictions, we designed and characterized 47 additional pathway variants to navigate its activity space, find optimal expression regions with desired activity response curves, and relieve rate-limiting steps in metabolism. Creating sequence-expression-activity maps accelerates the optimization of many protein systems and allows previous measurements to quantitatively inform future designs.

**Keywords** biophysical models; pathway optimization; SEAMAPs; synthetic biology

**Subject Categories** Methods and Resources; Synthetic Biology and Biotechnology

**Mol Syst Biol. (2014) 10: 731**

## Introduction

A microbe's ability to sense its environment, process signals, perform decision making, and manufacture chemical products is ultimately controlled by its DNA sequence (Yim _et al_, 2011; Du _et al_, 2012, 2013; Moon _et al_, 2012; Sandoval _et al_, 2012; Santos _et al_, 2012; Tseng & Prather, 2012; Lee _et al_, 2013; Xu _et al_, 2013; Zhao _et al_, 2013). The genetic part sequences controlling protein expression levels directly affect an organism's behavior by modulating binding occupancies, rates of catalysis, and the competition for shared resources. Finding a quantitative relationship between DNA sequence and host behavior has been a central goal toward understanding evolution and adaptation, treating human disease, and the engineering of organisms for biotechnology applications (Strohman, 2002; Wessely _et al_, 2011; O'Brien _et al_, 2013; de Vos _et al_, 2013; Quandt _et al_, 2014).

Recent advances in DNA synthesis, assembly, and mutagenesis have greatly accelerated the construction and modification of large synthetic genetic systems. Combinatorial assembly methods enable the simultaneous introduction and modification of genetic parts to create many genetic system variants (Engler _et al_, 2009; Gibson _et al_, 2009; Guye _et al_, 2013; de Raad _et al_, 2013; Sleight & Sauro, 2013; Coussement _et al_, 2014; Dharmadi _et al_, 2014; Torella _et al_, 2014). The development of multiplex genome engineering provides the ability to simultaneously introduce DNA mutations into several genomic loci (Wang _et al_, 2009, 2012; Esvelt & Wang, 2013). Further, Cas9-dependent and TALE-dependent nicking, cleavage, and mutagenesis have expanded site-directed genome engineering to diverse organisms (Miller _et al_, 2010; Bassett _et al_, 2013; Chang _et al_, 2013; Lo _et al_, 2013; Mali _et al_, 2013; Ran _et al_, 2013). Although techniques are readily available to construct or modify large genetic systems of interest, we currently cannot predict the DNA sequences that will achieve an optimal behavior, particularly when the actions of multiple proteins are responsible for a system's function.

This design challenge could be solved by creating a quantitative link between a genetic system's sequence, protein expression levels, and behavior in order to predict the effects of DNA mutations, map the phenotypic space accessible by natural evolution, and optimize non-natural DNA sequences toward a desired genetic system performance. These relationships, called sequence-expression-activity maps (SEAMAPs), can be formulated by combining sequence-dependent models predicting changes in protein expression levels (Rhodius & Mutalik, 2010; Salis, 2011; Brewster _et al_, 2012; Johnson _et al_, 2012; Borujeni _et al_, 2013; Kilpinen _et al_, 2013) with system-level models describing genetic system function (Fendt _et al_, 2010;

1  Department of Chemical Engineering, Pennsylvania State University, University Park, PA, USA
2  Department of Biological Engineering, Pennsylvania State University, University Park, PA, USA
   *Corresponding author. Tel: +1 814 865 1931; E-mail: salis@psu.edu

Sneppen *et al*, 2010; Hyeon & Thirumalai, 2011; Wessely *et al*, 2011; Smallbone *et al*, 2013). Several types of models can be combined, utilizing thermodynamics, kinetics, mass transfer, and dynamical systems theory, to describe the multi-scale physical interactions affecting system function (Hyeon & Thirumalai, 2011). Validation of SEAMAP predictions across the range of possible behaviors critically tests our knowledge of the system's interactions, the modeling assumptions, and provides a systematic approach to optimizing genetic system function toward the most desired behavior (Smolke & Silver, 2011; Yadav *et al*, 2012).

However, the development and parameterization of such sequence-expression-activity models has been stymied by several forms of combinatorial explosion (Bailey, 1999; Alper & Stephanopoulos, 2004; Zelcbuch *et al*, 2013). First, the number of regulatory genetic part sequences that differentially control protein expression is astronomical; a single 30-nucleotide genetic part has $10^{18}$ possible sequences, while five of these genetic parts have more possible sequences than atoms in the universe. Second, while protein expression can be modulated over a $> 100,000$-fold range, there are relatively few genetic part sequences that express an extremely high or low amount of protein; finding these sequences using anecdotal rules or random mutagenesis is unreliable. Third, when multiple proteins synergistically work together to achieve the best possible behavior, the probability of finding genetic part sequences with optimal protein expression levels decreases combinatorially. Fourth, the differences in gene expression machineries may cause a genetic system to function well in one organism, but not another. New approaches to developing SEAMAPs will be needed to circumvent combinatorial explosion, particularly as larger genetic systems with many proteins are targeted for engineering. Notably, while system-level models describing protein interactions have several unknown parameters, model reduction and rule-based simulations can significantly reduce the number of equations, transitions, co-dependent variables, and insensitive constants (Conzelmann *et al*, 2008; Tran *et al*, 2008; Apgar *et al*, 2010; Sneddon *et al*, 2010).

In this work, we demonstrate that biophysical modeling and computational design can be combined to efficiently create predictive sequence-expression-activity models for multi-protein genetic systems, while circumventing combinatorial explosion (Fig 1A). First, we employ predictive biophysical models to map the relationship between sequence and expression and to develop an automated search algorithm that rationally designs the smallest number of genetic system variants whose protein expression levels cover the largest portion of the multi-protein expression space. This design process is tailored for each genetic system with the goal of maximizing the observable changes in system behavior across the entire physiologically possible range. Second, we characterize a small number of genetic system variants and use overall activity measurements to parameterize a system-level mechanistic model that predicts how changes in protein expression control system function. The resulting SEAMAP for the genetic system is repeatedly used to correctly design and optimize its function with different targeted behaviors.

We first extensively validate our search algorithm's ability to design maximally informative genetic system variants using 646 genetic system variants, encoded on both plasmids and genomes, and characterized in diverse gram-negative and gram-positive bacteria. We then demonstrate that characterizing a small number of these maximally informative genetic system variants is sufficient to

map the system's multi-dimensional expression space. We carry out our sequence-expression-activity mapping approach on an prototypical 3-enzyme carotenoid biosynthesis pathway, characterizing only 73 pathway variants to build a SEAMAP that predicts all possible physiological behaviors of the pathway. We then test the model's predictions by correctly designing 19 additional pathway variants to access intermediate activities (interpolation); 28 additional pathway variants to access higher activities, including optimal pathway variants (extrapolation); and transcriptionally regulated pathway variants with desired activity response curves. We also use SEAMAP predictions to understand the relationship between DNA mutations and the pathway's evolutionary landscape. Finally, we compare three types of system-level models (mechanistic, geometric, and statistical) to analyze their ability to design genetic systems with targeted activities and to provide re-usable design information.

## Results

### Efficient searching of the sequence-expression space

The first step to mapping a genetic system's sequence, expression, and activity relationship is to design sequences to efficiently search its expression space. We developed an automated search algorithm, called the RBS Library Calculator, to design the smallest synthetic ribosome-binding site (RBS) library that systematically increases a protein's expression level across a selected range on a $> 10,000$-fold proportional scale. To do this, we use a predictive biophysical model to map mRNA sequence to translation initiation rate, employing a previously developed, and recently expanded, statistical thermodynamic model of the bacterial ribosome's interactions. We combine this sequence-dependent model with a genetic algorithm to perform optimization. Through iterations of *in silico* mutation, recombination, prediction, and selection, synthetic RBS library sequences using the 16-letter degenerate alphabet are designed to maximize the search coverage of a selected translation rate space, while minimizing the number of RBS variants in the library (Fig 1B). When incorporated into well-designed operons (see Box 1 for design rules), the proteins' expression levels will be proportional to their translation initiation rates.

The algorithm has several modes to navigate multi-protein expression spaces. These modes enable one to control the search range, search resolution, and sequence design constraints according to the application and design objectives. In Search mode, a synthetic RBS library is optimized to cover the widest possible expression space using a desired resolution. In Genome Editing mode, the synthetic RBS library is designed toward introducing the fewest, consecutive mutations into the genome. Finally, in Zoom mode, the translation rate range is narrowed, and the search resolution is increased, to design an RBS library to target optimal expression levels.

To validate the RBS Library Calculator's Search mode, three optimized RBS libraries were designed using high, medium, or low search resolutions with 36, 16, or 8 variants per library, respectively, to control reporter protein expression on a multi-copy plasmid in *Escherichia coli* DH10B. Degenerate RBS sequences primarily utilized 2-nucleotide degeneracies (S, K, R, B, and M) with only one instance of a 3-nucleotide degeneracy (M). None contained

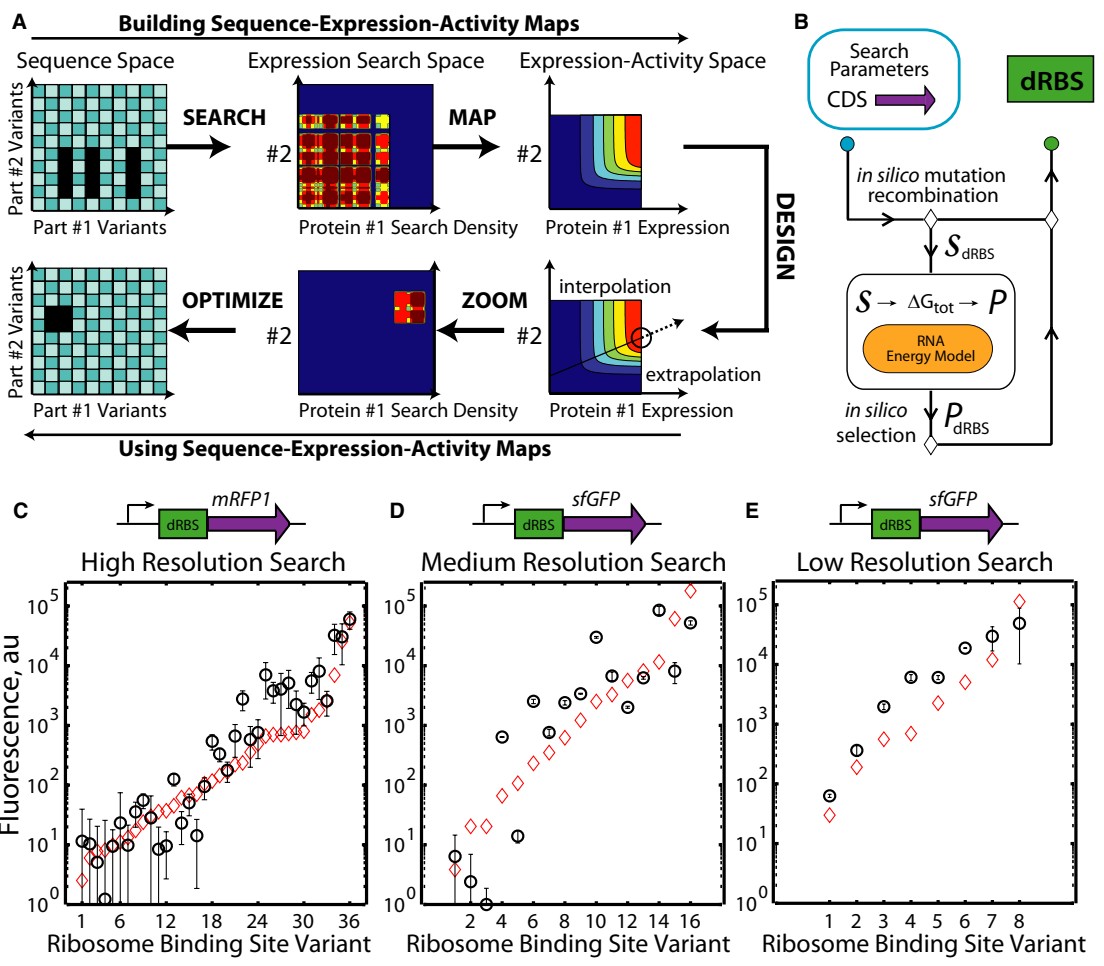

**Figure 1.  Building and using sequence-expression-activity maps (SEAMAPs).**

A    The RBS Library Calculator designs a synthetic RBS library to efficiently search a multi-dimensional protein expression space. A kinetic mechanistic model maps the relationship between protein expression levels and genetic system activity, using a minimal number of measurements for parameterization. The SEAMAP's predictions are repeatedly used for different design objectives. The RBS Library Calculator designs a new RBS library to zoom onto a region of targeted protein expression levels for optimized genetic system performance.

B    The RBS Library Calculator combines a biophysical model of translation with a genetic algorithm to identify the smallest degenerate RBS sequence (dRBS) with maximal search coverage for an input protein-coding sequence (CDS). The biophysical model calculates the ribosome's binding free energy $\Delta G_{tot}$ for an input mRNA sequence $S$, which is then related to its translation initiation rate and protein expression level $P$.

C–E  Fluorescence measurements show that optimized RBS libraries in *Escherichia coli* DH10B searched a 1-dimensional expression level space with 94, 79, and 99% search coverages at high, medium, and low search resolutions, respectively. Translation initiation rate predictions (red diamonds) are compared to measurements (Pearson $R^2$ is 0.88, 0.79, and 0.89, respectively. All *P*-values < 0.001). Data averages and standard deviations from 6 measurements.

a 4-nucleotide degeneracy (N). Search mode inserted degenerate nucleotides 5–19 upstream of the start codon to modulate both the 16S rRNA-binding affinity and the unfolding energetics of inhibitory mRNA structures. We quantified the optimized RBS libraries' search ranges, coverages, and translation rate predictions by measuring reporter protein expression levels from individual RBS variants within each library. Fluorescence measurements were taken during 24-h cultures maintained in the early exponential growth phase by serial dilutions to achieve steady-state conditions. All DNA sequences, translation rate predictions, and fluorescence measurements are provided in the Supplementary Table S1.

Fluorescence measurements show that the optimized RBS libraries searched the 1-dimensional (1 protein) expression level spaces with high coverages, high dynamic ranges, and accurate translation rate

predictions. The 36-variant RBS library systematically increased *mRFP1* expression from low to high levels with a 49,000-fold dynamic range and 94% search coverage (Fig 1C), while the 16-variant RBS library uniformly increased *sfGFP* expression across a 84,000-fold range with only a small coverage gap at 100 au (79% search coverage) (Fig 1D). The lowest resolution RBS library contained only eight variants, but uniformly increased *sfGFP* expression between the selected translation rate range, yielding protein expression levels from 63 to 49,000 au (778-fold dynamic range) with a high 99% search coverage (Fig 1E).

The biophysical model of bacterial translation accurately predicted the translation initiation rates from these 60 RBS variants with an average error $\Delta\Delta G_{total}$ of 1.74 kcal/mol, which is equivalent to predicting the measured translation initiation rate to within

**Box 1:    Design rules for synthetic bacterial operons**

Protein expression levels are affected by several factors, including transcription rates, mRNA stabilities, and translation rates. Proportional control over expression is achieved by optimizing RBS libraries to vary translation initiation rates, while carrying out rational sequence design to minimize changes in other factors. Codon usages are optimized to increase their translation elongation rates, while reducing the number of internal start codons with high translation initiation rates (Quan *et al*, 2011). Changes in mRNA stability are reduced by shortening unprotected mRNA regions and by removing potential RNAse-binding sites, including long single-stranded or duplexed RNA regions (Saito & Richardson, 1981; Dasgupta *et al*, 1998; Baker & Mackie, 2003; Folichon *et al*, 2003). Intergenic regions are designed to limit the extent of translational coupling within multicistronic bacterial operons (Oppenheim & Yanofsky, 1980). Using these rules, the effects of confounding control variables are minimized, thereby enabling designed RBSs to proportionally alter a protein's expression level, regardless of its location within a bacterial operon. Importantly, building and using SEAMAPs employs a reference genetic system variant; its predictions depend only on proportional changes to protein expression.

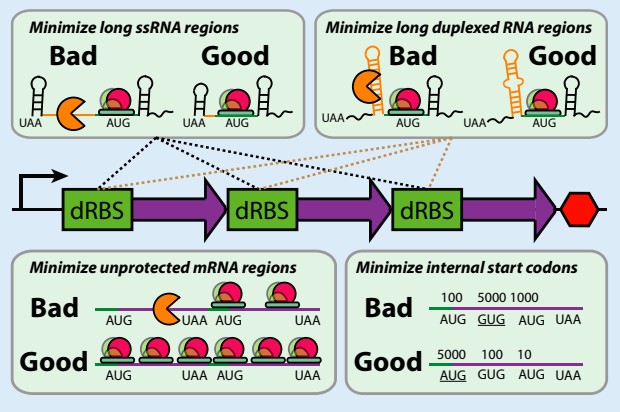

2.2-fold with an apparent β of 0.45 mol/kcal. The biophysical model's predictions were particularly accurate for the high- and low-resolution libraries (average $\Delta\Delta G_{total}$ = 1.05 kcal/mol, $R^2$ = 0.88; and average $\Delta\Delta G_{total}$ = 0.46 kcal/mol, $R^2$ = 0.89, respectively) in contrast to the medium-resolution library that contains several outliers at low expression levels (average $\Delta\Delta G_{total}$ = 3.71 kcal/mol, $R^2$ = 0.72).

**Navigation of expression spaces in diverse bacterial species**

The ribosome's interactions with mRNA can change between organisms, leading to varying translation rates. By accounting for known differences, we investigated whether the biophysical model can accurately predict translation rates in diverse bacterial hosts. Using the model to predict expression differences between organisms would enable the development of host-independent SEAMAPs and the engineering of genetic systems in one organism for their eventual use in another. We selected four bacterial hosts currently used for biotechnology applications (Supplementary Table S2): *E. coli* BL21 for overexpression of recombinant proteins; *Pseudomonas fluorescens* for production of biopolymers and soil decontamination; *Salmonella typhimurium* LT2 for secretion of large proteins, including spider silk; and *Corynebacterium*

*glutamicum* for production of enzymes and amino acids (Monti *et al*, 2005; Widmaier *et al*, 2009; Becker *et al*, 2011). To predict translation initiation rates, we included promoter-dependent upstream sequences in the 5′ UTR and selected the appropriate 3′ 16S rRNA sequence for each host, 5′- ACCUCCUUU-3′ for the gram-positive *C. glutamicum,* and 5′-ACCUCCUUA-3′ for the remaining gram-negative species (Fig 2A).

We designed a 16-variant-optimized RBS library to vary mRFP1 expression across a 14,000-fold range, introduced expression cassettes into broad host vectors with host-specific promoters, and measured fluorescence during long-time cultures in host-specific media (Materials and Methods). Overall, fluorescence measurements varied between 1,051- and 10,900-fold, depending on the host, and show that relative translation initiation rates were correctly predicted to within 2.1-fold (average error $\Delta\Delta G_{total}$ of 1.61 kcal/mol, Pearson $R^2$ is 0.89) (Fig 2B). Translation rate predictions were more accurate in gram-negative *E. coli* BL21 ($\Delta\Delta G_{total}$ of 1.18 kcal/mol, Pearson $R^2$ is 0.93), compared to gram-positive *C. glutamicum* ($\Delta\Delta G_{total}$ of 1.81 kcal/mol, Pearson $R^2$ is 0.88), though the difference is a single outlier (Supplementary Table S1). Interestingly, the apparent Boltzmann constant used to convert calculated binding free energies into predicted translation rates did not significantly vary between bacterial hosts (apparent β was 0.42 ± 0.02 mol/kcal). Consequently, these observations suggest that the free energies of *in vivo* RNA–RNA interactions remain the same regardless of the host organism, including the effects of molecular crowding on binding events (Tan *et al*, 2013).

**Efficient search in gram-positive and gram-negative bacterial genomes**

Genome engineering techniques enable the targeted mutagenesis of genomic DNA, either by employing oligo-mediated allelic recombination, homologous recombination, or site-directed non-homologous end joining (Sharan *et al*, 2009; Wang *et al*, 2009; Urnov *et al*, 2010; Cho *et al*, 2013; Cong *et al*, 2013; Esvelt & Wang, 2013; Mali *et al*, 2013). We developed the Genome Editing mode to identify the minimal number of adjacent genomic RBS mutations that systematically increase a protein's expression level across a wide range. Optimization is initialized using the wild-type genomic RBS and protein-coding sequences, and the solution is used to perform genome mutagenesis. We then evaluated our ability to search expression spaces by using optimized RBS libraries to modify the genomes of gram-positive and gram-negative bacteria.

First, we employed homologous recombination to introduce an optimized library of heterologous cassettes into the *Bacillus subtilis* 168 genome, using Genome Editing mode to optimize two RBS libraries that control expression of the reporter *mRFP1* with translation initiation rates from 100 to 96,000 au on the model's proportional scale. Translation rate predictions use 5′-ACCUCCUUU-3′ as the 3′ end of the *B. subtilis* 16S rRNA. Fluorescence measurements of 14 single clones from the libraries show that single-copy *mRFP1* expression varied from 10 and 17,600 au with a search coverage of 76%, well-predicted translation initiation rates that were proportional to the measured expression levels ($R^2$ = 0.81), and with a low error in the calculated ribosomal interactions (average $\Delta\Delta G_{total}$ = 1.77 kcal/mol) (Fig 3A).

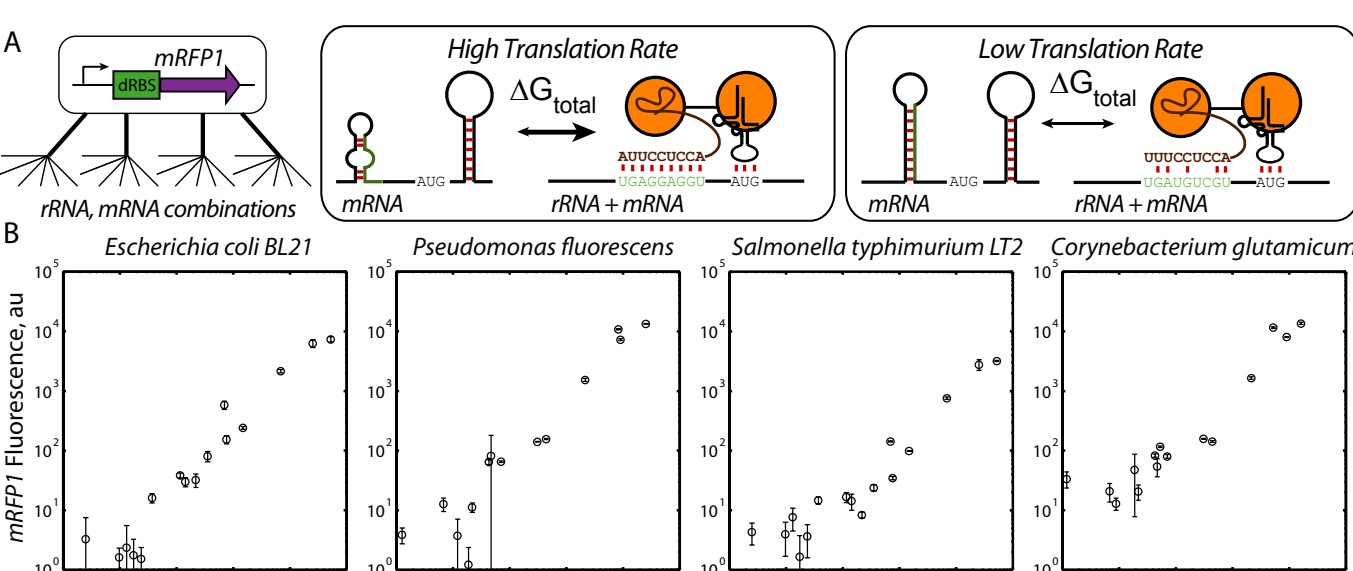

**Figure 2. Searching expression spaces in diverse bacterial hosts.**

A  Differences in bacterial 16S rRNA sequences lead to different mRNA translation rates. To overcome combinatorial explosion, the biophysical model predicts how changes in mRNA and rRNA sequences control translation rate.

B  A 14- to 16-variant-optimized RBS library controlling *mRFP1* expression was characterized in *Escherichia coli* BL21, *Pseudomonas fluorescens*, *Salmonella typhimurium* LT2, and *Corynebacterium glutamicum*. The biophysical model accurately predicted translation initiation rates across a > 1,000-fold range in *E. coli* BL21 ($\Delta\Delta G_{total}$ is 1.18 kcal/mol, $R^2$ is 0.93), *P. fluorescens* ($\Delta\Delta G_{total}$ is 1.63 kcal/mol, $R^2$ is 0.90), *S. typhimurium* LT2 ($\Delta\Delta G_{total}$ is 1.83 kcal/mol, $R^2$ is 0.89), and *C. glutamicum* ($\Delta\Delta G_{total}$ is 1.81 kcal/mol, $R^2$ is 0.88). All *P*-values $< 10^{-6}$. Data averages and standard deviations from three measurements.

Second, we employed MAGE mutagenesis on the *E. coli* MG1655-derived *EcNR2* genome (Wang *et al*, 2009), targeting its *lacI-lacZYA* locus and controlling *lacZ* protein expression levels (Fig 3B). We first conducted three rounds of MAGE mutagenesis to introduce an in-frame stop codon into the *lacI* repressor-coding sequence (Supplementary Table S3). Using the algorithm's Genome Editing mode, we then designed a 12-variant degenerate oligo-nucleotide to target the *lacZ* RBS sequence and uniformly increase its translation initiation rate from 20 to 55,000 au. We conducted twenty rounds of MAGE mutagenesis to introduce the 12 sets of RBS mutations into the genome and selected 16 colonies for sequencing of the *lacZ* genomic region. Ten of these colonies harbored genomes with unique mutated RBS sequences controlling *lacZ* translation. *lacZ* activities from the derivative EcNR2 genomes were individually measured using Miller assays after long-time cultures maintained in the early exponential growth phase (Fig 3B). The measured *lacZ* expression levels varied across a 2,400-fold range, searched the expression space with 84% coverage, and were well predicted by the biophysical model's predicted translation initiation rates up to 3,000 au on the model's proportional scale ($R^2 = 0.93$).

Interestingly, increasing the *lacZ* translation initiation rate beyond 3,000 au, which is four-fold over its wild-type rate, did not further increase *lacZ* activity, suggesting that there is a critical point where translation initiation may no longer be the rate-limiting step in protein expression, potentially due to ribosomal pausing during translation elongation, or protein misfolding. Significant changes in specific growth did not occur (Supplementary Table S1). To further increase expression, one could replace the existing, natural genes in the genome with newly designed protein-coding sequences optimized for maximum expression control, which motivates the design of synthetic genomes (Lajoie *et al*, 2013).

### Efficient search in multi-dimensional expression spaces

Complex genetic systems express multiple proteins to carry out their function. Building multi-protein SEAMAPs are particularly difficult, as it requires searching a larger, multi-dimensional expression space. We next evaluated Search mode's ability to efficiently explore a 3-dimensional expression space, compared to several types of randomly generated RBS libraries. Optimized RBS libraries were designed by the RBS Library Calculator to encode 8 RBS variants with predicted translation initiation rates across a 5,000-fold range (Supplementary Table S4). They contained 2-nucleotide degeneracies at distributed positions from 4 to 26 nucleotides upstream of the start codon, including positions far from the Shine-Dalgarno sequence. Separately, we constructed random RBS libraries by selecting six nucleotides of the Shine-Dalgarno sequence and randomly incorporating all possible choices to create 4,096 variants with widely different predicted translation initiation rates (Supplementary Table S5). We employ combinatorial DNA assembly to construct bacterial operon variants encoding *cfp*, *mRFP1*, and *gfpmut3b* reporter proteins (Gibson *et al*, 2009), generating either 512 operon variants when using optimized RBS libraries, and up to 68.7 billion operon variants when using random RBS libraries. The extent of DNA library assembly is limited, and only a sub-sample of the randomized bacterial operon variants will ever be constructed or selected for characterization.

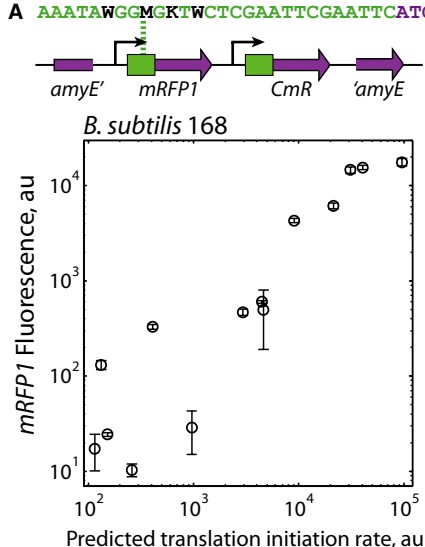

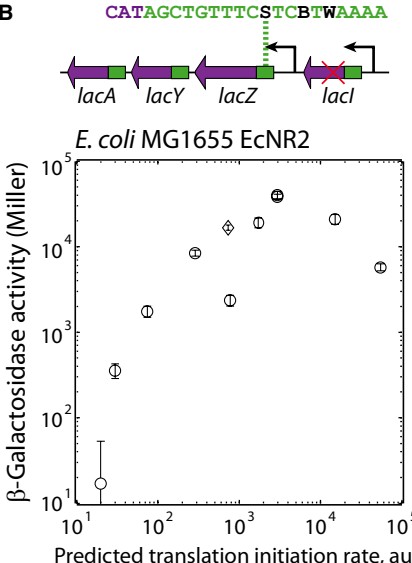

**Figure 3.   Searching genomic expression spaces.**

A   Two RBS libraries were optimized to control the expression of a genomic single copy of *mRFP1*, incorporated into the *amyE* locus of *Bacillus subtilis*. Fluorescence measurements from 14 clones were compared to their predicted translation initiation rates (Pearson $R^2$ is 0.81, P-value is $2 \times 10^{-6}$). The expression space was searched with 76% coverage. Data averages and standard deviations from three measurements.

B   A 12-variant RBS library was optimized to control genomic *lacZ* expression. Predicted translation initiation rates are compared to measured *lacZ* activities (circles), including the wild type (diamond), showing a linear relationship below the activity plateau (Pearson $R^2$ is 0.93, P-value is 0.02). The expression space was searched with 84% coverage. Data averages and standard deviations from four measurements.

We compared search coverages when using either optimized or randomly mutagenized RBS libraries. For each case, 500 strains with operon variants were randomly selected and individually cultured, and their CFP, mRFP1, and GFPmut3b fluorescences were quantified by flow cytometry. The optimized RBS libraries searched the 3-dimensional protein expression level space across a 20,000-fold

range with a 42% search coverage (Supplementary Fig S1). In contrast, the randomly mutagenized RBS library only partly covered the expression level space, showing a high degree of clustering that is responsible for decreasing its search coverage to 14% (Supplementary Fig S2), which agrees well with the computationally predicted search coverage of 14.7% using Monte Carlo sampling of the predicted translation initiation rates (Supplementary Methods). Using the same method, similar search coverages of 16.7 and 19.1% are computationally predicted for a 4,096-variant library NNNGGANNN (Mutalik *et al*, 2013) and a 23,328-variant library DDRRRRRDDDD (Wang *et al*, 2009), respectively. Overall, a minority of randomly generated operon variants expressed higher or lower levels that would be necessary for many applications. Using the algorithm's Search mode, higher-dimensional expression spaces may be efficiently sampled with high coverages at targeted resolutions (Supplementary Fig S3 and Supplementary Tables S15, S16 and S17).

**Mapping the sequence-expression-activity space of a multi-enzyme pathway**

To demonstrate our approach, we created a SEAMAP for a multi-enzyme biosynthesis pathway and then repeatedly used it to optimize the pathway's sequences and expression levels for different design objectives. The RBS Library Calculator in Search mode was employed to systematically vary *crtEBI* enzyme expression levels originating from a carotenoid biosynthesis pathway in *Rhodobacter sphaeroides* and codon-optimized for *E. coli*. Three 16-variant-optimized RBS libraries were designed to vary *crtE*, *crtB*, and *crtI* from 445 to 72,000 au, 3 to 20,000 au, and 97 to 203,000 au, respectively (Supplementary Table S6). Three-part combinatorial DNA assembly onto a ColE1 vector resulted in up to 4,096 clonal pathway variants, transcribed by the arabinose-induced $P_{BAD}$ promoter. Seventy-three clones containing unique pathway variants were randomly selected, sequenced, transformed into *E. coli* MG1655-derived EcHW2f strain (Supplementary Table S3), and cultured for a 7-h post-induction period. Their neurosporene contents were determined by hot acetone extraction and spectrophotometry. Within a single library, the pathways' neurosporene productivities uniformly varied between 3.3 and 196 µg/gDCW/h (Fig 4A and Supplementary Table S7).

Using optimized RBS libraries yielded a large continuum of pathway activities with the smallest number of measurements. Biophysical model predictions from sequenced RBSs indicate that the translation rates broadly explored the selected 3-dimensional space (Fig 4B), which eliminates redundant measurements and thus maximizes the measurements' information content. As *crtEBI* translation rates were increased, pathway productivities did not reach a plateau, suggesting that translation initiation remained the rate-limiting step throughout the mapped space.

To formulate an expression-activity relationship for the pathway, we developed a mechanistic, kinetic model to describe the system-level behavior (Materials and Methods). We listed the 24 elementary chemical reactions that are responsible for enzymatic conversion of isoprenoid precursors (DMAPP and IPP) to neuro-sporene. All reactions are reversible, including enzymes' binding to substrates and the release of products (Fig 4D and Supplementary Fig S4). We developed a kinetic model of the reaction network, deriving a system of differential equations with 48 unknown parameters. Mole balances on each enzyme and flux constraints reduced the

system of equations to having 33 unknown parameters (Supplementary Methods). We then use an ensemble modeling approach (Tran *et al*, 2008; Contador *et al*, 2009) that combines model reduction and dimensional analysis to compare the pathway variants' calculated fluxes to a reference pathway, and to convert simulated reaction fluxes to measurable productivities. Changing a pathway variant's translation rates proportionally controls the simulated enzyme concentrations, which alters the pathway's predicted neurosporene productivity. Finally, we employed model identification to determine a unique set of kinetic model parameters that reproduced the measured neurosporene productivities for the 72 non-reference pathway variants, across ten independent and randomly initialized optimization runs (Supplementary Table S8). The resulting kinetic model maps *crtEBI* translation rates to neurosporene productivities across a 10,000-fold, 3-dimensional translation rate space (Fig 4B).

## Design and optimization of multi-enzyme pathways using SEAMAPs

We tested the SEAMAP's predictions by using it to design *crtEBI* pathway variants according to different design objectives, beginning with pathways that exhibit intermediate activities (interpolation). Nineteen additional pathway variants were characterized, and the pathway's predicted productivities were compared to measured productivities. The biophysical model predicts the *crtEBI* translation rates from sequenced RBSs, and the kinetic model uses these translation rates to calculate each pathway variant's productivity (Fig 4E). The kinetic model correctly determined how changing the enzymes' translation rates controlled the pathway's productivity (24% error across a 100-fold productivity range) (Supplementary Table S9). Overall, kinetic model predictions were more accurate at higher *crtEBI* translation rates (Supplementary Fig S5). In general, a high *crtE* translation rate was necessary for high biosynthesis rates, while low *crtB* and high *crtI* translation rates were sufficient to balance the pathway.

Second, we tested the SEAMAP's ability to design improved pathways by identifying an expression region with higher activities beyond the existing observations (extrapolation) and employing the RBS Library Calculator in Zoom mode to target this region. We designed 8-variant RBS libraries with translation rate ranges of 32,000–305,000 au for *crtE*, 1,800–232,000 au for *crtB*, and 26,000–1,347,000 au for *crtI* (Fig 4C and Supplementary Table S10). After combinatorial DNA assembly, 28 clones containing unique pathway variants were randomly selected, sequenced, and cultured for a 7-h post-induction period. The resulting neurosporene productivities improved up to 286 μg/gDCW/h (Fig 4A) (Supplementary Table S11). Importantly, the SEAMAP predicts sequence-dependent changes to pathway productivity, but not the effects of media formulation or growth conditions. For example, the best pathway variant's productivity was further increased to 441 μg/gDCW/h when optimized media and aeration conditions were employed (Alper *et al*, 2006) (Supplementary Fig S6).

Third, we tested the SEAMAP's ability to predict a pathway's activity response curve when utilizing a tunable promoter to vary the *crtEBI* operon's transcription rate (Fig 5A). Changes in the transcription rate will proportionally vary all enzyme expression levels in the *crtEBI* operon, causing diagonal shifts in expression

space that will lead to productivity changes. We modified the most productive *crtEBI* pathway variant, replacing its $P_{BAD}$ promoter with the $P_{lacO1}$ promoter, and measured a productivity of 119 μg/gDCW/h in supplemented M9 media without IPTG addition. This measurement is used as a reference point, identifying where the modified pathway variant is located in the sequence-expression-activity space (Fig 5B, black square). The same promoter replacement was performed for three additional pathway variants with distinctly different *crtEBI* translation rates (Supplementary Table S12). Under these growth conditions, the change in promoter resulted in an overall increase in enzyme expression levels.

Model calculations are then combined to show how changes in the promoter's transcription rate and the pathway variants' translation rates will affect their productivities (Fig 5B). We use a model to first relate IPTG concentration to $P_{lacO1}$ transcription rate (Supplementary Fig S7), followed by multiplication with the translation rates to determine enzyme expression levels. These enzyme expression levels are substituted into the SEAMAP to predict the pathways' productivities. According to the model, the pathway variants' productivities will increase with promoter induction, up to a maximum amount (Fig 5C, left). The maximum productivity is determined by the pathway variants' translation rate ratios and the promoter's transcription rate. Notably, the model predicts that excess enzyme expression will lower a pathway's productivity, due to sequestration of intermediate metabolites as enzyme–substrate complex.

We then characterized the four pathway variants' productivities with increasing IPTG induction (Supplementary Table S12). Though the pathway variants were expressed by the same promoter, their activity responses varied greatly and were highly consistent with model calculations (Fig 5C, right). Pathway variants with optimal translation rates had higher productivities and achieved maximum activity at a lower transcription rate. However, additional increases in transcription lowered their productivities, due to excess enzyme expression levels. The SEAMAP shows how changing the operon's transcription and translation rates can exhibit this nonlinear behavior. Consequently, one can use the SEAMAP to guide the selection of a regulated promoter to dynamically control a pathway's activity. Regulated promoters can often serve as sensors for cellular stress, and they may be used to implement feedback control over a pathway's enzyme expression levels to maintain maximal activities. The use of dynamic regulation has been shown to significantly improve a pathway's productivity (Zhang *et al*, 2012; Dahl *et al*, 2013).

## The expanding search for optimal pathways

Analysis of the *crtEBI* pathway's SEAMAP reveals why metabolic optimization efforts have been generally laborious. First, each enzyme has the potential to be a rate-limiting step in the pathway (Bailey, 1999). Distributed control over the pathway's flux requires that all enzyme expression levels must be tuned to achieve high productivities. In particular, large changes in enzyme expression levels are needed to exert control; small changes in enzyme levels are buffered by compensating changes in metabolite concentrations (Fendt *et al*, 2010). This principle illustrates the need for genetic parts that maximally change protein expression levels across a wide range. Second, although the goal of

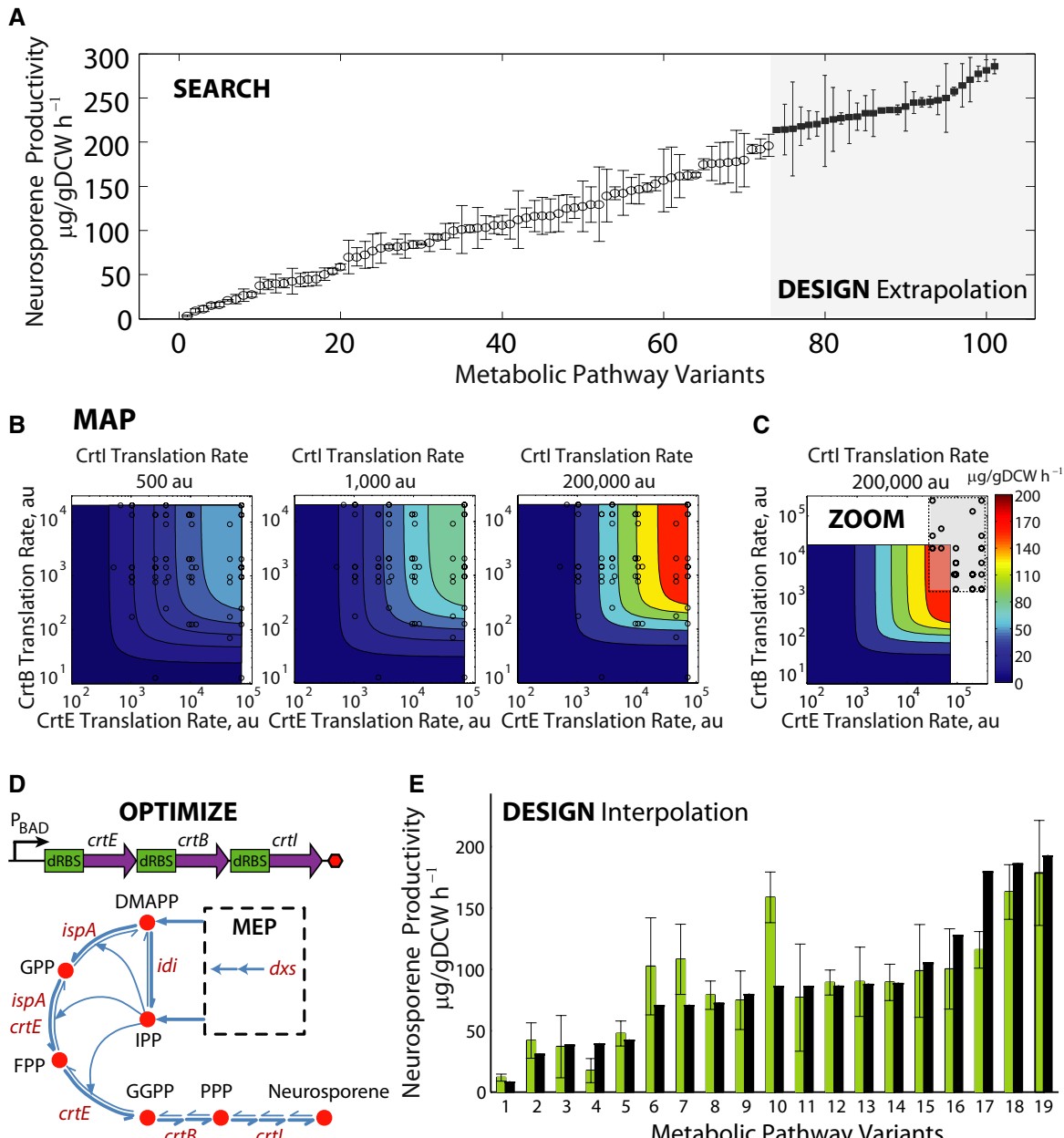

**Figure 4. Sequence-expression-activity mapping of a multi-enzyme pathway.**

A Characterization of two libraries of neurosporene biosynthesis pathway variants, using the RBS Library Calculator for searching a large combinatorial space (Search mode, left) or a narrow targeted region (Zoom mode, right). Averages and standard deviations from at least three measurements of neurosporene productivities.

B Measurement data and translation rate predictions (circles) from Search mode are used to parameterize a kinetic model of the pathway's reaction rates, showing the relationship between *crtEBI* translation rates and neurosporene productivity.

C To design pathways with higher activities, a translation rate region (gray box) is targeted using the RBS Library Calculator in Zoom mode. Translation rate predictions from selected pathway variants are shown (circles).

D A schematic of the bacterial operon-encoding *crtEBI*, and the corresponding reactions, genes, and metabolites in the biosynthesis pathway. Cofactors are not shown.

E To evaluate the design of pathways with intermediate activities, 19 additional crtEBI pathway variants were characterized, and the predicted neurosporene productivities (black bars) were compared to the measured productivities (green bars). Data averages and standard deviations from two measurements.

pathway optimization is to continually increase pathway productivities, it remains unclear when an engineered pathway has reached its maximum productivity, and thus, pathway engineering efforts continue until a better variant cannot be found. A quantitative criteria for pathway optimality would provide a metric for

when to expand metabolic engineering efforts to additional proteins and pathways. Next, we use SEAMAP predictions to calculate a pathway variant's optimality, and to determine when a pathway variant has become optimally balanced. We then demonstrate that reaching this optimality condition is the

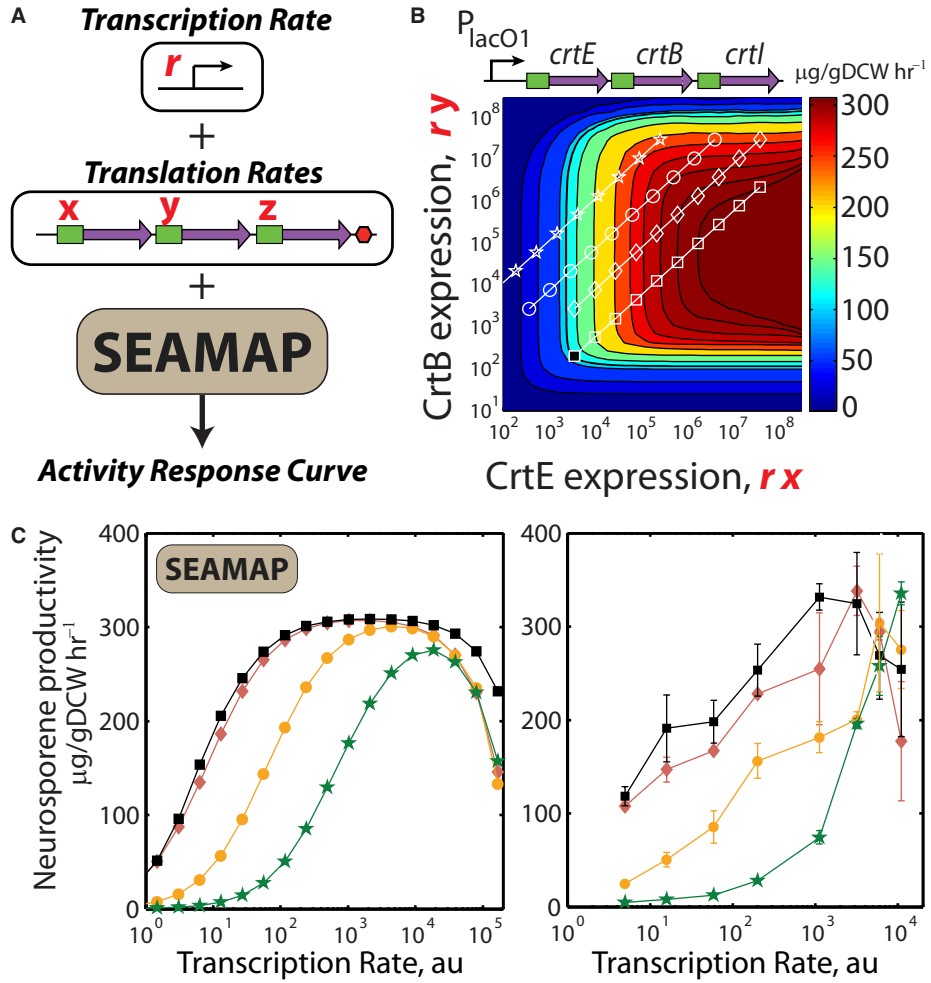

**Figure 5.  Using SEAMAPs to design multi-enzyme pathways with desired activity response curves.**

A  A promoter's transcription rate *r* and the *crtEBI* translation rates (*x*, *y*, *z*) are inputted into SEAMAP's kinetic model to determine the pathway's productivity.

B  A slice of the CrtEB expression-activity space is shown, where CrtI expression is 200,000 au. The effects of transcriptional regulation for four pathway variants are shown as diagonal lines at their respective translation rates. The productivity of a reference pathway variant in one condition (black square) was characterized to determine its location in expression-activity space, which provides orientation for all other locations.

C  Left: The effect of the P$_{lacO1}$ promoter's transcription rate on the pathway variants' productivities is calculated. The location of the global maxima depends on the promoter's transcription rate and the mRNA translation rates. Right: The productivities of the four pathway variants are measured as transcription rate is increased via IPTG induction. Changes in translation rate cause the global maxima to appear at lower transcription rates, consistent with model calculations. Data averages and standard deviations from two measurements.

appropriate time to redirect metabolic engineering efforts and increase precursor biosynthesis rates.

We define pathway optimality using flux control coefficients (FCCs) (Fell, 1992; Kholodenko & Westerhoff, 1993) and use the SEAMAP predictions to calculate the FCCs for the *crtEBI* pathway. FCCs quantify how differential fold changes in enzyme expression control a pathway's overall productivity and vary depending on the enzymes' expression levels (Fig 6A; Supplementary Fig S8). High FCC regions indicate where increasing an enzyme's expression will increase pathway's productivity, while low FCC regions show where increasing expression does not lead to a significant improvement in productivity. Negative FCC regions show where excess enzyme expression causes metabolite sequestration or growth toxicity.

A pathway is *balanced* when its enzymes' FCCs are equal; differential fold increases in enzyme expression all have the same effect on pathway activity. Further, a pathway is *optimally balanced* when its enzymes' FCCs are zero at the global maxima in activity space; increasing enzyme expression has a minimal, or negative, impact on pathway productivity. Once a pathway is optimally balanced, it has shifted activity control over to the upstream metabolic module responsible for precursor biosynthesis.

We next investigated whether a pathway variant's FCCs provide the stopping criterion that indicates the need for further metabolic engineering of upstream pathways. We selected an optimally balanced pathway variant where all of its enzymes have approximately zero FCCs and an imbalanced pathway variant where a positive FCC for *crtE* indicates that it remains a rate-limiting step (0.65 for CrtE, near 0 for CrtB and CrtI) (Fig 6A). We then employed the RBS Library Calculator's Genome Editing mode to optimize a 16-variant RBS library controlling genomic *dxs*

expression, the enzyme that controls the first committed step to isoprenoid biosynthesis (Supplementary Table S13). Using co-selection MAGE mutagenesis (Wang *et al*, 2012), 16 genome variants were constructed and verified. The RBS library varied *dxs* translation from 110 to 291000 au. The increase in *dxs* expression significantly improved the optimally balanced pathway's productivity up to 517 µg/gDCW/h (Fig 6B; Supplementary Table S14), while only increasing the imbalanced pathway's productivity up to 81 µg/gDCW/h (Fig 6C). Thus, a pathway variant's flux control coefficients provided the quantitative criteria for indicating when to cease pathway engineering efforts and redirect toward improving precursor biosynthesis rates. These results suggest an iterative metabolic engineering strategy where upstream pathways are additionally optimized, applying the optimality criteria in a successive fashion.

More broadly, both the optimization of genetic systems and the study of evolutionary dynamics can be understood as time-iterated DNA mutations to navigate an organism's sequence-expression-activity space. Evolution acts on a slower time scale and only selects for population members whose activities have improved their overall fitness; in contrast, optimizing genetic systems can be conducted independent of fitness evaluations and can be directed toward rare DNA sequences. Both processes operate on the same sequence-expression-activity landscape. Using the multi-enzyme pathway's SEAMAP, we show how evolution could shape pathway productivity, due to random mutation. Starting from the optimally balanced pathway shown in Fig 6, the effects of single, double, and triple RBS mutations are calculated, showing that pathway productivity decreases in almost all cases (Fig 7). From an evolutionary perspective, if the multi-enzyme pathway is essential to cell growth, then these mutations will never proliferate in the population. However, for pathways involved in manufacturing chemical products, pathway productivity is more likely to be inversely coupled to cell growth, which will lead to mutation enrichment. Thus, building

SEAMAPs allows one to visualize a genetic system's evolutionary landscape and potentially to design DNA sequences toward becoming insensitive to evolutionary dynamics.

## Discussion

A key challenge to successfully engineering cellular organisms has been the combinatorial vastness of their genetic instruction space, and the complex relationship between genotype and phenotype. We present a new approach to overcoming this design challenge by combining a biophysical model of gene expression with a system-level mechanistic model to quantitatively connect a genetic system's sequence, protein expression levels, and behavior. We illustrate how to efficiently build sequence-expression-activity maps (SEAMAPs), performing the fewest number of characterization experiments, by using an automated search algorithm to uniformly explore a multi-dimensional expression space (Fig 1). Both the biophysical model and automated search algorithm are highly versatile; they can be used in diverse gram-negative and gram-positive bacteria, and to modify both plasmids and genomes (Figs 2 and 3). Using the search algorithm, we built a SEAMAP for a multi-enzyme pathway and demonstrated how it can be used to design pathway variants with targeted productivities (Fig 4) and tailored activity response curves (Fig 5), while quantitatively guiding further metabolic engineering efforts to increase precursor biosynthesis rates (Fig 6). Altogether, creating a SEAMAP for a genetic system provides a coherent and predictive model that can be repeatedly used to optimize non-natural sequences and achieve complex design objectives.

Overall, our computational design approach combines principles from both systems and synthetic biology to build predictive models (Kitano, 2002). We formulate models using physical principles that can be re-used across different systems and scales (Hyeon &

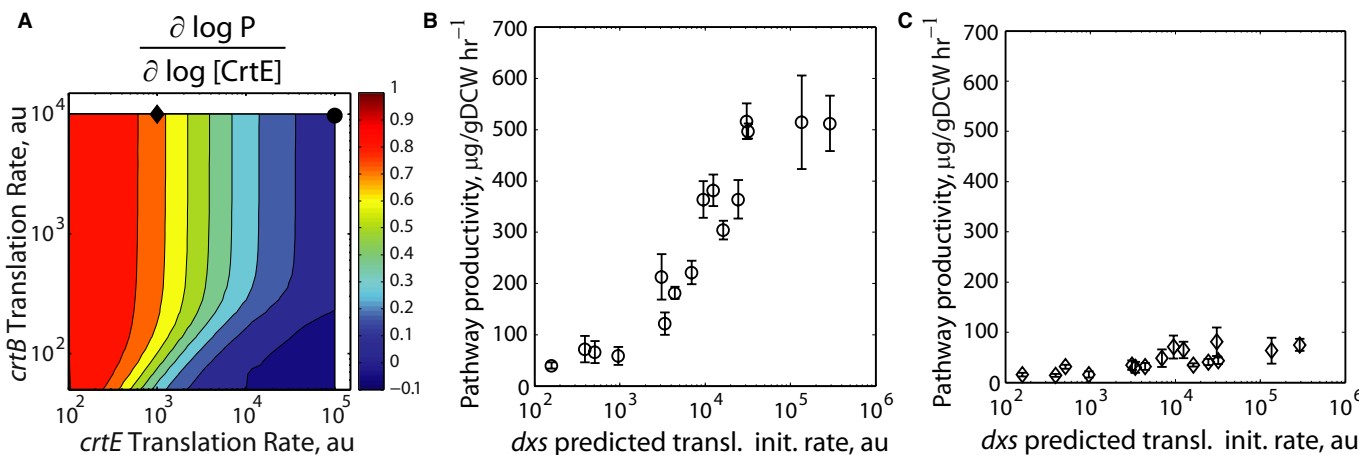

**Figure 6.  Increasing precursor biosynthesis for optimally balanced versus imbalanced pathways.**

A    The relationship between *crtEB* translation rates and CrtE's flux control coefficient (FCC) is calculated using SEAMAP predictions. A lower FCC indicates that the enzyme is less rate limiting. Here, the *crtI* translation rate is 100,000 au. According to their FCCs, increasing precursor biosynthesis is predicted to improve the optimally balanced pathway variant (black circle) more than the imbalanced pathway variant (black diamond).

B, C    An optimized RBS library is integrated to control genomic *dxs* translation initiation rate and systematically vary precursor biosynthesis, followed by productivity measurements using either (B) an optimally balanced pathway or (C) an imbalanced pathway variant. Predicted *crtEBI* translation initiation rates are (305,000 au; 17,120 au; 886,364 au) for the optimally balanced pathway variant and (1,046 au; 20,496 au; 200,300 au) for the imbalanced pathway variant.

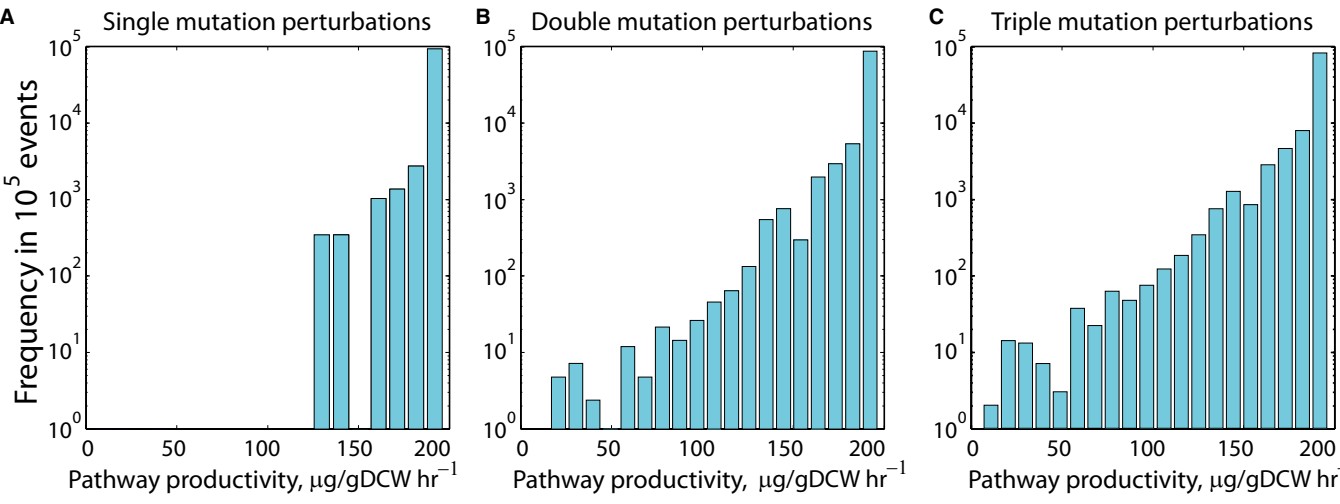

**Figure 7. Predicting the evolutionary landscape of a multi-enzyme pathway.**
Histograms show that random mutations will more likely decrease a pathway's productivity.

A–C    Either (A) one-, (B) two-, or (C) three-nucleotide mutations are randomly introduced into the 35 nucleotide-long ribosome-binding site sequences of the *crtEBI* operon. Changes in enzyme expression levels and pathway productivities are predicted using the SEAMAP.

Thirumalai, 2011), and their unknown parameters are parameterized using rationally designed genetic system variants with synthesized sequences. The number of unknown parameters can scale proportionally with the number of proteins in a genetic system, particularly in biosynthesis pathways and engineered genetic circuits where proteins typically interact with a small number of partners. Specifically, modeling each reversible reaction in a multi-enzyme pathway requires two kinetic parameters, and modeling a transcription factor's binding occupancy in a genetic circuit requires one thermodynamic parameter. Our search algorithm greatly improves the efficiency of model building by designing the minimal number of genetic system variants that maximally span the multi-dimensional expression-activity space, eliminating redundant measurements and increasing the information content of datasets. As a result, the parameterized model can accurately represent the genetic system's nonlinear behavior over its sequence-expression-activity space, while reducing characterization effort. These aspects of our approach become more important as larger genetic systems are modified and characterized, for example, using chip-based oligo-nucleotide synthesis and next-generation sequencing (Goodman *et al*, 2013; Kosuri *et al*, 2013).

When mechanistic information is unavailable, there are black-box approaches to building expression-activity relationships that utilize geometry, informatics, and statistics. Using our pathway variant dataset as an example, we illustrate the advantages and limitations of two alternative non-mechanistic system-level models. First, we use computational geometry to decompose the four-dimensional expression-activity space into Voronoi polygons, and to calculate unknown activities using linear interpolation between adjacent Voronoi cells (Supplementary Information). Using this approach, a geometric SEAMAP could use intermediate enzyme expression levels to predict pathway activities with 15% error (interpolation) (Supplementary Fig S9, Supplementary Fig S10). However, to design pathway variants with higher enzyme expression levels and activities (extrapolation), we would need to simplify the expression-activity relationship outside the model's convex hull, for example, by assuming linearity and independence. Based on the mechanistic modeling and FCC calculations, we know such assumptions are incorrect; the expression-activity relationship is nonlinear, and all enzyme expression levels co-dependently control pathway flux. Consequently, geometric models can be used to formulate accurate expression-activity models, although their ability to design genetic systems outside the previously characterized expression space is limited.

Second, we employ a statistical linear regression model to relate expression to activity, using an Exterior Derivative Estimator to determine best-fit coefficients (Aswani *et al*, 2011) that is similar to a recent approach (Lee *et al*, 2013). This statistical model assigns RBS variants as categories that are present or absent for each gene and finds coefficients that linearly relate individual RBS variants' effects on expression to measured pathway productivities. The linear regression model could predict the pathway variants' activities with a 46% error (Supplementary Information). Using a previous approach (Lee *et al*, 2013), when the productivity data are log-transformed, a statistical model with the same training size predicted the pathways' activities with a 10% error. To design new pathway variants, the category-based statistical model predicts pathway productivities when utilizing RBS variants that were characterized in the existing training set. However, it cannot predict pathway productivities when utilizing newly designed RBS variants, which creates gaps in the expression-activity relationship. This becomes important when, for example, the optimal expression levels to achieve a desired design objective fall within a gap in the characterized expression space, or are located outside the initially selected expression range; a newly designed RBS sequence would be needed to achieve optimality. Additional information on these non-mechanistic modeling approaches and their differences are available in the Supplementary Information and Supplementary Table S18.

Regardless of the modeling approach, it is important to critically test whether parameterization using randomly generated expression-activity data also leads to an accurate representation.

A well-defined model should not equally represent both real and random data. We carried out a test of this null hypothesis and found that the kinetic, computational geometry, and linear regression models all had high prediction errors (70–110%), except when log-transforming the random productivity data (15%). The variable transformation allowed the random data to fit the statistical model with similar error, compared to the real data (Supplementary Fig S11). These comparisons provide an important baseline to error measurements.

Our approach to designing genetic systems also departs from recent efforts to characterize genetic part toolboxes to control expression (Mutalik *et al*, 2013). A part-centric approach to a statically tuning expression requires a tremendous amount of characterization to ensure that parts are modular, orthogonal, and insensitive to surrounding sequence changes, while being similarly functional in diverse hosts (Nielsen *et al*, 2013). Further, to be useful, a part-centric toolbox must both be wide and deep. Many genetic parts with similar expression levels must be available, together with many genetic parts that span the entire expression level range. Genetic parts must not have long repetitive sequences to minimize rates of homologous recombination and undesired navigation of the sequence-expression-activity space (Lovett, 2004; Sleight *et al*, 2010). By combining sequence-dependent biophysical models with optimization, we can generate an unlimited number of non-repetitive genetic parts that span the entire expression range, and in diverse bacterial hosts. In another distinction, a toolbox is a static list of DNA sequences and cannot incorporate additional design criteria *ex post facto* without additional characterization to ensure similar function. However, as demonstrated, our validated model can be repeatedly re-used with different design objectives to optimize a genetic system's sequence toward a desired behavior.

As synthetic biology matures into an engineering discipline, additional effort will be needed to ensure continuity between design approaches, where numbers have a physical meaning and the origins of design rules have a molecular basis. In that regard, our computational design approach is seamlessly augmented by an improved understanding of translation initiation (Borujeni *et al*, 2013) and the development of new mechanistic models that use sequence information to predict changes in DNA bending and looping, transcription factor and nucleosome binding, transcription initiation and elongation rate, transcriptional termination efficiency, RNA folding and stability, polarity, translation elongation and coupling, regulation by cis- and trans- RNAs, and other interactions that affect protein expression levels (Geggier & Vologodskii, 2010; Rhodius & Mutalik, 2010; Brewster *et al*, 2012; Johnson *et al*, 2012; Chen *et al*, 2013; Kilpinen *et al*, 2013; Rodrigo *et al*, 2013). The integration of models, to relate sequence to expression and expression to activity, should be a central goal of synthetic biology as it will expand our ability to navigate a genetic system's behavior space, and to optimize entire synthetic genomes to achieve a desired sensing, signaling, and metabolic performance.

A software implementation of the RBS Library Calculator is available at http://www.salis.psu.edu/software, online since 2011. As of April 2014, 463 unaffiliated researchers have designed 4,354 optimized RBS libraries for diverse biotechnology applications. The algorithm's predictions have already been independently verified by a non-affiliated laboratory (Coussement *et al*, 2014).

## Materials and Methods

### Strains and plasmid construction

All strains and plasmids are listed in Supplementary Table S2.

To construct plasmid-based RBS libraries in *E. coli* strain DH10B, protein-coding sequences (mRFP1 or sfGFP) were PCR amplified from pFTV1 or pFTV2 using mixed primers that encode optimized degenerate RBSs. The gel-purified PCR product was joined with digested, gel-purified vector backbone using a 2-part chew-back anneal-repair (CBAR) reaction (Gibson *et al*, 2009) to create the pIF1, pIF2, and pIF3 expression plasmids. Plasmids were transformed into *E. coli* DH10B, selected on chloramphenicol, and verified by sequencing. Expression plasmids contain a ColE1 origin of replication, a chloramphenicol resistance marker, the J23100 sigma$^{70}$ constitutive promoter, the optimized degenerate ribosome-binding site, and the selected reporter gene. Selected plasmids from the pIF1 series were transformed into *E. coli* BL21 and *S. typhimurium* for expression characterization. mRFP1 expression cassettes from the same plasmids were sub-cloned into a modified pSEVA351 vector, replacing J23100 with a Ptac promoter (GenBank Accession JX560335, CmR, OriT replication origin), and transformed into *P. fluorescens* and *C. glutamicum* B-2784 by electroporation.

To construct genomic RBS libraries in *B. subtilis* strain 168, a *Bacillus* integration vector pDG1661 was modified by replacing the spoVG-*lacZ* region with an mRFP expression cassette, containing the pVeg constitutive promoter from *Bacillus*, an RBS sequence flanked by BamHI and EcoRI restriction sites, the *mRFP1*-coding sequence, and a T1 terminator. A mixture of annealed oligonucleotides containing optimized RBS libraries was inserted between the BamHI and EcoRI sites by ligation, and the constructs were verified by sequencing. The integration vector was integrated into the *amyE* genomic locus of *B. subtilis 168* using the standard protocol, selected on 5 μg/ml chloramphenicol, and the integration verified by iodide starch plate assay.

To construct genomic RBS libraries in *E. coli* EcNR2 (Wang *et al*, 2009), 90mer oligonucleotides were designed to have minimal secondary structure at their 5′ and 3′ ends and were synthesized with 5′ phosphorothioate modifications and 2′ flurouracil to improve their allelic replacement efficiencies (Integrated DNA Technologies, Coralville, Iowa). Their concentrations were adjusted to 1 uM in water. The EcNR2 strain was incubated overnight in LB broth with antibiotic (50 μg/ml ampicillin or chloramphenicol) at 30°C and with 200 RPM orbital shaking. The culture was then diluted to early exponential growth phase (OD$_{600}$ = 0.01) in 5 ml SOC, reaching mid-exponential growth phase within 2–3 h. When reaching an OD$_{600}$ of 0.5–0.7, the culture was warmed to 42°C for 20 min and then placed on ice. One milliliter culture was centrifuged for 30 s at >10,000 *g*, and the supernatant was discarded. The cell pellet was washed twice with chilled water, dissolved in the oligo aqueous solution, and electroporated using an Eppendorf electroporator (model 2,510) at 1,800 V. The culture was recovered by incubation in pre-warmed SOC at 37°C until reaching an OD$_{600}$ of 0.5–0.7. The culture was then used for an additional cycle of mutagenesis, plated on LB agar to obtain isogenic clones, or pelleted to make glycerol stocks. Mutagenesis was verified by sequencing PCR amplicons of the *lacZ* locus.

To vary genomic *dxs* expression, co-selection MAGE was performed on an *E. coli* EcNR2 strain whose *lacZ* contains two early stop codons, performing 12 mutagenesis cycles using 1 µM of an oligonucleotide mixture to introduce the *dxs* RBS library and 10 nM of an oligonucleotide to restore *lacZ* expression (Supplementary Table S2). Culturing and selection took place over a 36-h growth period in M9 minimal media supplemented with 0.4% lactose at 30°C and shaking at 250 RPM. Forty colonies were selected, and 16 unique RBS variants were verified by sequencing PCR amplicons.

To combinatorially assemble 3-reporter operons in *E. coli* strain DH10B, PCR amplicons containing Cerulean, mRFP1, and GFPmut3b/vector backbone were amplified from pFTV3 using mixed primers containing optimized degenerate RBS sequences and 40 bp overlap regions. The PCR products were Dpn1 digested, gel-purified, and joined together into the pFTV vector using a 3-part CBAR assembly reaction (Gibson *et al*, 2009), using the existing J23100 constitutive promoter. The library of plasmids was transformed into *E. coli* DH10B and selected on LB plates with 50 µg/ml chloramphenicol.

To combinatorially assemble *crtEBI* operons driven by a $P_{BAD}$ promoter, the crtE-coding sequence was first sub-cloned into a FTV3-derived vector that replaced the constitutive J23100 promoter with an *araC*-$P_{BAD}$ cassette, followed by PCR amplification of crtE, crtB, and crtI/vector using mixed primers containing optimized degenerate RBS sequences and 40 bp overlap regions. PCR products were joined together using a 3-part CBAR assembly reaction to create a library of plasmids, which was transformed into *E. coli* DH10B, selected on LB agar plates with 50 µg/ml chloramphenicol. Isolated pathway variants were verified by sequencing. *crtEBI*-coding sequences originated from *Rhodobacter sphaeroides* 2.4.1 and were codon-optimized and synthesized by DNA 2.0 (Menlo Park, CA).

**Growth and measurements**

*Escherichia coli* strains and *P. fluorescens* were cultured in LB broth Miller (10 g tryptone, 5 g yeast extract, 10 g NaCl) or M9 minimal media (6 g $Na_2HPO_4$, 3 g $KH_2PO_4$, 1 g $NH_4Cl$, 0.5 g NaCl, 0.24 g $MgSO_4$, 0.011 g $CaCl_2$), as indicated. *Salmonella typhimurium* LT2 and *C. glutamicum* B-2784 were cultured in LB Lennox broth (10 g tryptone, 5 g yeast extract, 5 g NaCl) and Brain Heart Infusion broth (6 g brain heart infusion, 6 g peptic digest of animal tissue, 14.5 g digested gelatin, 3 g glucose, 5 g NaCl, 2.5 g $Na_2HPO_4$, 7.4 pH), respectively.

To record fluorescence measurements from RBS variants controlling reporter expression, transformed strains and a wild-type *E. coli* DH10B strain were individually incubated overnight at 37°C, 200 RPM in a 96-deep-well plate containing 750 µl LB broth and 50 µg/ml chloramphenicol, or 50 µg/ml streptomycin for the DH10B strain. Five microliter of the overnight culture was diluted into 195 µl M9 minimal media supplemented with 0.4 g/l glucose, 50 mg/l leucine, and 10 µg/ml antibiotic in a 96-well micro-titer plate. The plate was incubated in a M1000 spectrophotometer (TECAN) at 37°C until its $OD_{600}$ reached 0.20. Samples were extracted, followed by a 1:20 serial dilution of the culture into a second 96-well micro-titer plate containing fresh M9 minimal media. A third plate was inoculated and cultured in the same way to maintain cultures in the early exponential phase of growth for 24 h. The fluorescence distribution of 100,000 cells from culture samples was recorded by a LSR-II Fortessa flow cytometer (BD biosciences). Protein fluorescences were determined by taking fluorescence distributions' averages and subtracting average auto-fluorescence. Growth temperature for *S. typhimurium* LT2, *P. fluorescens*, and *C. glutamicum* was 30°C.

To record fluorescence measurements from 3-reporter operon libraries, 500 colonies were randomly selected and grown individually using LB Miller media with 50 µg/ml chloramphenicol, for 16 h at 37°C with 200 RPM orbital shaking, inside a 96-deep-well plate. Cultures were then diluted 1:20 into fresh supplemented LB Miller media within a 96-well micro-titer plate, incubated at 37°C in a M1000 spectrophotometer (TECAN) until the maximum $OD_{600}$ reached 0.20. The blue, red, and green fluorescence distributions of samples were recorded using flow cytometry, applying a previously calibrated color correction to remove cross-fluorescence. The average blue, red, and green fluorescence is determined by subtracting average DH10B auto-fluorescence.

To record *lacZ* activities using Miller assays, *E. coli* EcNR2 genome variants containing *lacI* knockouts and *lacZ* RBS mutations were grown overnight at 30°C with 250 RPM orbital shaking in a 96-deep-well plate containing LB Miller and 50 µg/ml chloramphenicol. Cultures were then diluted into fresh supplemented LB Miller media and cultured at 30°C to an $OD_{600}$ of 0.20. Twenty microliter of cultures were diluted into 80 µl permeabilization solution and incubated at 30°C for 30 min. Twenty-five microliter samples were then transferred into a new microplate to perform Miller assays. One hundred and fifty microliter of ONPG solution was added, and absorbances at 420 and 550 were recorded by the M1000 for a 3 h period. Using this data, Miller units were calculated by finding the average value of ($OD_{420}$ − 1.75 $OD_{550}$)/$OD_{600}$ during the times when the product synthesis rate was constant.

To measure neurosporene productivities, pathway variants were incubated for 16 h at 30°C, 250 RPM orbital shaking in 5 ml culture tubes, then washed with PBS, dissolved in fresh LB miller (50 µg/ml chloramphenicol, and 10 mM arabinose), and grown for another 7 h. Cells were centrifuged (Allegra X15R at 4,750 RPM) for 5 min, washed with 1 ml ddH2O, and dissolved in 1 ml acetone. The samples were incubated at 55°C for 20 min with intermittent vortexing, centrifuged for 5 min, and the supernatants transferred to fresh tubes. Absorbance was measured at 470 nm using NanoDrop 2000c spectrophotometer and converted to µg neurosporene (× 3.43 µg/nm absorbance). The remaining pellet was heated at 60°C for 48 h to determine dry cell weight. Neurosporene content was calculated by normalizing neurosporene production by dry cell weight. Neurosporene productivity was determined by dividing by 7 h.

To record neurosporene productivity under optimized growth conditions, pathway variants were incubated overnight in 5 ml LB miller, followed by inoculating a 50-ml shake flask culture using 2×M9 media supplemented with 0.4% glucose and 10 mM arabinose. The culture was grown for 10 h at 37°C with 300 RPM orbital shaking. The neurosporene productivity was measured using 10 ml of the final culture as stated above. To record neurosporene productivity from pathway variants using IPTG-inducible promoters, cultures were grown overnight and then diluted into 50 ml 2×M9 media supplemented with 2% glucose, grown at 30°C with

250 RPM shaking, and induced with increasing IPTG concentrations. Pathway productivity was recorded after 22 h of growth.

## Models and Computation

### The RBS calculator

The RBS Calculator v1.1 was employed to calculate the ribosome's binding free energy to bacterial mRNA sequences, and to predict the translation initiation rate of a protein-coding sequence on a proportional scale that ranges from 0.1 to 100,000 or more. The thermodynamic model uses a 5-term Gibbs free energy model to quantify the strengths of the molecular interactions between the 30S ribosomal pre-initiation complex and the mRNA region surrounding a start codon. The free energy model is:

$$\Delta G_{total} = \Delta G_{mRNA:rRNA} + \Delta G_{spacing} + \Delta G_{start} + \Delta G_{standby} - \Delta G_{mRNA} \quad (1)$$

Using statistical thermodynamics and assuming chemical equilibrium between the pool of free 30S ribosomes and mRNAs inside the cell, the total Gibbs free energy change is related to a protein-coding sequence's translation initiation rate, $r$, according to:

$$r \propto \exp\left(-\beta \Delta G_{total}\right) \quad (2)$$

This relationship has been previously validated on 132 mRNA sequences where the $\Delta G_{total}$ varied from −10 to 16 kcal/mol, resulting in well-predicted translation rates that varied by over 100,000-fold (Salis *et al*, 2009). The apparent Boltzmann constant, $\beta$, has been measured as $0.45 \pm 0.05$ mol/kcal, which was confirmed in a second study (Hao *et al*, 2011). In practice, we use a proportional constant of 2,500 to generate a proportional scale where physiological common translation initiation rates vary between 1 and 100,000 au.

In the initial state, the mRNA exists in a structured conformation, where its free energy of folding is $\Delta G_{mRNA}$ ($\Delta G_{mRNA}$ is negative). After assembly of the 30S ribosomal subunit, the last nine nucleotides of its 16S rRNA have hybridized to the mRNA while all non-clashing mRNA structures are allowed to fold. The free energy of folding for this mRNA–rRNA complex is $\Delta G_{mRNA:rRNA}$ ($\Delta G_{mRNA:rRNA}$ is negative). mRNA structures that impede 16S rRNA hybridization or overlap with the ribosome footprint remain unfolded in the final state. These Gibbs free energies are calculated using a semi-empirical free energy model of RNA and RNA–RNA interactions (Xia *et al*, 1998; Mathews *et al*, 1999) and the minimization algorithms available in the Vienna RNA suite, version 1.8.5 (Gruber *et al*, 2008).

Three additional interactions will alter the translation initiation rate. The tRNA$^{fMET}$ anti-codon loop hybridizes to the start codon ($\Delta G_{start}$ is most negative for AUG and GUG). The 30S ribosomal subunit prefers a five-nucleotide distance between the 16S rRNA-binding site and the start codon; non-optimal distances cause conformational distortion and lead to an energetic binding penalty. This relationship between the ribosome's distortion penalty ($\Delta G_{spacing} > 0$) and nucleotide distance was systematically measured. Finally, the 5′ UTR binds to the ribosomal platform with a free energy penalty $\Delta G_{standby}$.

There are key differences between the first version of the RBS Calculator (v1.0) (Salis *et al*, 2009) and version v1.1 (Salis, 2011).

The algorithm's use of free energy minimization was modified to more accurately determine the 16S rRNA-binding site and its aligned spacing, particularly on mRNAs with non-canonical Shine-Dalgarno sequences, and to accurately determine the unfolding free energies of mRNA structures located within a protein-coding sequence. For the purpose of this work, a ribosome-binding site (RBS) sequence is defined as the 35 nucleotides located before the start codon of a protein-coding sequence within a mRNA transcript. However, the presence of long, highly structured 5′ UTRs can further alter the translation initiation rate of a protein-coding sequence by manipulating its $\Delta G_{standby}$. The ribosome's rules for binding to long, highly structured 5′ UTRs has been characterized (Espah Borujeni *et al*, 2014) and will be incorporated into a future version of the RBS Calculator (v2.0).

### The RBS library calculator

The objective of the RBS Library Calculator is to identify the smallest RBS library that uniformly varies a selected protein's expression level across a targeted range to efficiently identify optimal protein expression levels and quantify expression-activity relationships. The RBS Library Calculator designs degenerate ribosome-binding site (RBS) sequences that satisfy the following mini-max criteria: First, the RBS sequence variants in the library shall express a targeted protein to maximize coverage, $C$, of the translation rate space between a user-selected minimum ($r_{min}$) and maximum rate ($r_{max}$); second, the number of RBS variants in the library, $N_{variants}$, shall be minimized. The allowable range of translation rates is between 0.10 au and over 5,000,000 au though the feasible minimum and maximum rates will also depend on the selected protein-coding sequence. These criteria are quantified by the following objective function:

$$F = 10C - 0.02N_{variants} \quad (3)$$

The coverage of an RBS library is determined by first converting the translation rate space into a $\log_{10}$ scale and discretizing it into equal width bins. For this work, the bin width $W$ is called the search resolution as it ultimately defines how many RBS variants will be present in the optimized RBS library. The total number of bins is determined by the user-selected maximum and minimum translation rates and the search resolution $W$, while the RBS library coverage $C$ is determined by the ratio between filled bins and total bins, according to the following equations:

$$B_{total} = \left\lceil \frac{(r_{max}/r_{min})}{W} \right\rceil \quad C = \frac{B_{filled}}{B_{total}} \quad (4)$$

For example, there will be a total of 17 bins when using a search resolution $W$ of 0.30 and a translation rate space between 1.0 and 100,000 au. A bin at position $y$ in translation rate space will be filled when at least one RBS variant in the library has a predicted translation initiation rate that falls within the range $[y/10^W, y\,10^W]$. An RBS library's coverage is one when all translation rate bins are filled by at least one RBS variant. The objective function $F$ has a maximum value of $1 - 0.02\,B_{total}$, which is achieved when all bins are filled by a single RBS variant, yielding the most compact RBS library that expresses a protein with uniformly increasing translation rates.

The solution to the RBS Library Calculator optimization problem is a list of near-optimal degenerate ribosome-binding site sequences. A degenerate RBS is a 35-nucleotide sequence that uses the 16-letter IUPAC code to indicate whether one or more nucleotides shall be randomly incorporated at a particular sequence position. The alphabet defines the inclusion of either single nucleotides (A, G, C, U/T), double nucleotides (W, S, M, K, Y, B), triple nucleotides (D, H, V), or all four nucleotides (N) in each sequence position. $N_{variants}$ is determined by the number of sequence combinations according to these degeneracies.

Chemical synthesis of degenerate DNA sequences creates a mixture of DNA sequence variants, which are then incorporated into a natural or synthetic genetic system, either plasmid-encoded or chromosomally encoded. Chemical synthesis of the degenerate DNA oligonucleotides may introduce non-random bias in nucleotide frequency, due to differences in amidite substrate-binding affinities. The concentrations of manually mixed precursors can be adjusted to eliminate this bias.

Several properties of the RBS Library Calculator's mini-max optimization problem have influenced the selection of an appropriate optimization algorithm. First, the number of possible degenerate RBS sequences is very large ($16^{35}$), though many of these sequences will yield the same objective function. Further, the relationship between a degenerate RBS sequence and its library coverage is highly nonlinear and discontinuous. The addition of degeneracy to some nucleotide positions will greatly increase library coverage, whereas modifying other nucleotide positions has no effect on coverage. The nucleotide positions that affect the library coverage will typically include portions of the Shine-Dalgarno sequence, but also other positions that modulate the energetics of mRNA structures. The locations of mRNA structures will depend on the selected protein-coding sequence, which will significantly influence the optimal degenerate RBS sequence. Consequently, an evolutionary (stochastic) optimization algorithm was chosen to rapidly sample diverse sequence solutions and use mixing (recombination) to identify nucleotide positions that are most important to maximizing library coverage.

A genetic algorithm is employed to identify an optimal degenerate RBS sequence that maximizes the objective function, *F*. The procedure performs iterative rounds of *in silico* mutation, recombination, and selection on a population of degenerate RBS sequences to generate a new population with improved fitness (Fig 1B). First, a mutation operator is defined according to the following frequencies: (i) 40%, two degenerate sequences are recombined at a randomly selected junction; 15%, the degeneracy of a randomly selected nucleotide is increased; 15%, the degeneracy of a randomly selected degenerate nucleotide is decreased; 15%, a non-degenerate nucleotide is mutated to another non-degenerate nucleotide; 10%, the degenerate sequence is not modified (designated elites); or 5%, a new degenerate sequence is randomly generated. Second, one or two degenerate sequences in the population are randomly selected with probabilities proportional to their evaluated objective functions, a randomly selected mutation operator is performed on these degenerate sequences, and the results are carried forward into the new population. This process is repeated until the objective function for the most-fit sequence has reached the maximum value, the maximum objective function has not changed for a user-selected number of iterations, or when the total number of iterations has reached a user-selected maximum. The top five degenerate RBS sequences in the population are then returned, including the predicted translation initiation rates for each variant in the RBS library.

The genetic algorithm typically requires 50–100 iterations to identify optimal degenerate RBS sequences, starting from a population of randomly generated, non-degenerate RBS sequences. During the optimization procedure, the most common mutational trajectory is the broad expansion of sequence degeneracy toward maximizing coverage of the translation rate space, followed by targeted reduction of degeneracy to eliminate RBS variants with similar translation rates. The number of iterations is substantially reduced when a rationally designed RBS sequence is used as an initial condition, particularly when the selected maximum translation rate is over 10,000 au.

*Kinetic model formulation, transformation, and identification*

Mass action kinetics was utilized to formulate an ordinary differential equation (ODE) model to quantify the rates of production and consumption of the 24 metabolite, free enzyme, and bound enzyme species in the pathway's reaction network. A derivation is found in the Supplementary Information. The reaction network includes 10 reversible reactions catalyzed by Idi, IspA, CrtE, CrtB, and CrtI enzymes, including reversible binding of substrate to enzyme and reversible unbinding of product from enzyme (Supplementary Fig S4). IspA, CrtE, CrtB, and CrtI catalyze multiple reactions. These reactions convert intracellular isopentenyl diphosphate (IPP) and fimethylallyl diphosphate (DMAPP) to neurosporeneid. An additional five mole balances on intracellular enzyme were derived. There are 48 unknown kinetic parameters.

De-dimensionalization of the model was carried out by transforming all metabolite and enzyme concentrations into ratios, compared to the concentrations in a reference pathway variant. For example, the forward $v_{f1}$ and reverse $v_{r1}$ reaction rates for the binding of IPP to *idi* enzyme were multiplied and divided by the reference pathway's concentrations for IPP and free *idi* enzyme, yielding:

$$v_{f1} = \underbrace{\left( k_1 * [\text{IPP}]_{\text{ref}} * [\text{idi}]_{\text{ref}}^{\text{total}} \right)}_{\text{apparent kinetic parameter}} * \underbrace{\frac{[\text{IPP}]}{[\text{IPP}]_{\text{ref}}}}_{\substack{\text{metabolite} \\ \text{concentration} \\ \text{ratio}}} * \underbrace{\frac{[\text{idi}]^{\text{free}}}{[\text{idi}]_{\text{ref}}^{\text{total}}}}_{\substack{\text{enzyme} \\ \text{concentration} \\ \text{ratio}}}$$

$$v_{r1} = \underbrace{\left( k_{-1} * [CM1]_{\text{ref}} \right)}_{\text{apparent kinetic parameter}} * \underbrace{\frac{[CM1]}{[CM1]_{\text{ref}}}}_{\substack{\text{enzyme} \\ \text{concentration} \\ \text{ratio}}}$$

(5)

As a result, metabolite and enzyme concentration ratios are compared across pathway variants using dimensionless units. Accordingly, the total enzyme concentration ratios for each pathway variant were determined by comparing a pathway variant's translation rates to the reference pathway's translation rates. As an example, the *crtE* concentration ratio is:

$$\underbrace{\frac{[CrtE]^{\text{total}}}{[CrtE]_{\text{ref}}^{\text{total}}}}_{\substack{\text{enzyme} \\ \text{concentration} \\ \text{ratio}}} = \underbrace{\frac{\text{translation initation rate of crtE in a pathway variant}}{\text{translation initation rate of crtE in the reference pathway}}}_{\text{translation initation rate ratio}}$$

(6)

The choice of the reference pathway variant will alter the apparent kinetic parameter values, but it will not alter the solution to the ODEs; increases in the apparent kinetic parameters are compensated by decreases in the enzyme concentration ratios. The reference pathway (#53) has predicted translation initiation rates of 72,268, 20,496, and 203,462 au for *crtE*, *crtB*, and *crtI,* respectively.

Numerical integration of the transformed kinetic model is carried out using a stiff solver (ode23s, MATLAB) over a 7-h simulated time period to correspond to experimental conditions. The inputs into the kinetic model are the kinetic parameter values and the total enzyme concentration ratios. The resulting neurosporene production fluxes $r_p$ are related to measured neurosporene productivities by comparison to the reference pathway according to:

$$\underbrace{\frac{r_{p,i}}{r_{p,ref}}}_{\substack{\text{simulated}\\\text{production}\\\text{flux ratio}}} = \underbrace{\frac{\text{predicted neurosporene productivity of the i}^{\text{th}}\text{ pathway variant}}{\text{measured neurosporene productivity of the reference pathway}}}_{\text{pathway productivity ratio}}$$

(7)

The reference pathway has a neurosporene productivity of 196 ug/gDCW/h when grown in LB media (non-optimized growth conditions). Each pathway variant will have a different neurosporene production flux and predicted neurosporene productivity as a result of the different total enzyme concentrations, controlled by the *crtEBI* translation rates according to Equation 6. The kinetic parameters remain constant for all pathway variants.

Model reduction and identification were carried out to reduce the number of model degrees of freedom and to determine the kinetic parameter values that best reproduced the measured neurosporene productivities for the 73 pathway variants designed using Search mode. From the 48 unknown kinetic parameters, 10 non-independent parameters were eliminated, and an additional 5 were constrained using available biochemical data (Supplementary Information). A genetic algorithm was employed to identify the model's kinetic parameter values that best predicted the neurosporene productivities of the 72 non-reference pathway. On average, the resulting model predicts the neurosporene productivities to within 32% of the measurements (Supplementary Fig S5). We then performed inverse model reduction to determine the 48 kinetic parameter values that define the identified kinetic model (Supplementary Table S8). Model identification can be performed on the non-reduced model, though it would result in greater variability in best-fit kinetic parameters, longer optimization convergence times, and a requirement for more characterized pathway variants to achieve the same predictive error.

**Supplementary information** for this article is available online: http://msb.embopress.org

## Acknowledgements

We thank A.E. Borujeni, B. Pfleger, J. Torella, and A. Khodayari for valuable discussion; H. Wang and G. Church (Harvard University) for the gift of strains EcNR2 and EcHW2f; A. Demirci and T. Wood (Penn State) for the kind gift of strains *E. coli* BL21, *P. fluorescens*, and *C. glutamicum*; and the researchers who use the interactive website for their valuable feedback. This research was supported by the Office of Naval Research (N00014-13-1-0074), an NSF Career Award (CBET-1253641), a DARPA Young Faculty Award to H.M.S., and start-up funds provided by the Penn State Institute for the Energy and the Environment. M.G. and M.E. were supported by a NSF Research Experience for Undergraduates program. Computational resources provided by an Amazon AWS Research Grant.

## Author contributions

IF and HMS designed the study, developed the algorithm, analyzed results, and wrote the manuscript. IF, MK, JC, ME and MG conducted the experiments.

## Conflict of interest

The authors declare competing financial interests: H.M.S. is a founder of *De Novo* DNA.

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
