## [Review Process File · Molecular Systems Biology]

Efficient search, mapping, and optimization of multi-protein genetic systems in diverse bacteria

Iman Farasat, Manish Kushwaha, Jason Collens, Michael Easterbrook, Matthew Guido, Howard M. Salis

Corresponding author: Howard M. Salis, Pennsylvania State University

Review timeline:

Submission date:	01 November 2013
Editorial Decision:	04 December 2013
Revision received:	02 March 2014
Editorial Decision:	16 April 2014
Revision received:	27 April 2014
Accepted:	06 May 2014

Editor: Maria Polychronidou

Transaction Report:

1st Editorial Decision

04 December 2013

Thank you again for submitting your work to Molecular Systems Biology. We have now heard back from the three referees who agreed to evaluate your manuscript. As you will see from the reports below, the referees acknowledge that the presented approach is potentially useful for synthetic biology applications. However, they raise a series of concerns, which should be carefully addressed in a revision of the manuscript.

Without repeating all the points listed below, some of the more fundamental issues are the following:

- As reviewers #1 and #2 point out, additional data and controls would be required to better support the findings shown in Figures 1-3.
- The performance of the presented framework in higher dimension pathways should be discussed.
- The manuscript needs to be streamlined and carefully edited so that the main conclusions are clear and easily accessible to a broad audience.

On a more editorial level, we would like to encourage you to include the source data for the figures that show essential quantitative data. (Additional information is available in the "Guide for Authors" section in our website at http://msb.msubmit.net/html/msb_author_instructions.html#a3.4)

If you feel you can satisfactorily deal with these points and those listed by the referees, you may wish to submit a revised version of your manuscript. Please attach a covering letter giving details of the way in which you have handled each of the points raised by the referees. A revised manuscript will be once again subject to review and you probably understand that we can give you no guarantee at this stage that the eventual outcome will be favorable.

Reviewer #1:

General comments:

Farasat, I et al. combined the previously developed "Ribosome binding site calculator" method (Salis et al. Nature Biotech 2009) with a genetic optimization algorithm to design a library of ribosome binding sites (RBS) to explore a wide range of expression levels.

This method can operate in three modes with high, medium and low resolutions by varying the number of RBS sequences per bin and may reduce the probability of generating sequences that produce similar expression levels.

The authors show a few examples of this method including variation of the expression level of a single gene that is integrated into the genome in *B. subtilis* and *E. coli*, modulation of the expression levels of three genes in an operon and optimization of the neurosporene production in a three-enzyme metabolic pathway.

By using a combination of kinetic modeling and biophysical modeling the authors argue that the ability to design a library of RBS sequences with different translation efficiencies for a group of genes yields a better search and optimization methodology than using random libraries.

The main idea to use a computational model based on biophysical interactions to guide experimental design is interesting and important. Also, developing methods for rationally sampling the design space are needed that can yield more information content from minimal samples. Such rational design approaches have been successfully adopted in pharma/biotech industry (media optimization, biotransformation conditions etc), and yet to be demonstrated in genetic engineering/metabolic engineering fields.

Whereas optimization of the metabolic pathway is impressive and potentially useful (Fig. 4), we think the other examples (Fig. 1-3) are minor modifications to the original RBS calculator and these results do not stand by themselves due to a lack of sufficient data and controls (see specific comments). We think Fig. 4 is the notable contribution of this paper and the motivation is similar to the recent Lee, ME et al. NAR 2013 paper that engineered the violacein pathway using a statistical model. Since this seem the main contribution (though a number of other interesting claims are made though with less rigor) it would seem that some detailed comparison of the present results with the approaches taken by the Lee paper in yeast are in order.

Perhaps one topic to address is the relative merits of the "zoom" functionality or rather the learning from the original library. We think the regression approach in Lee is likely more efficient than the present one when the landscape is additive (non-epistatic) but the RBS library calculator approach may be a better method for optimizing systems with nonlinear production landscapes. The violacein pathway seems to be fairly linear in its response to expression (all main pathway components high seems to be pretty good). It would have been nice to see a more complete analysis of the neurosporene pathway parameter space. The utility of these approaches is deeply tied, we think, to the properties of these parameter spaces that map expression element activity to productivity for particular pathways.

Even if the pathway does have a complex topology to the parameter space, we suspect the proposed method by Farasat, I et al. does not scale well with the dimension of the pathway. For higher dimension systems, small modifications to the Lee method may be more useful. Again-- the interest here would be in the dissection of this issue.

Some of the figures of the present paper are very confusing and unclear (especially Fig. 1A and Fig. 4) with some extraneous details that are only tangentially related to the point (e.g. the reference to the game of "Battleship" in Fig. 1 may seem clever but it detracts from the point). In addition, overall we found that the writing is not very clear and could be improved (for example, explaining the relationship between the kinetic model and the RBS library calculator and providing sufficient detail to understand the figures).

Specific comments:

- Fig 1. Is confusing and can be simplified.
- The author's conclusions based on Fig. 2 and 3 are not supported by sufficient data and controls. In addition, important information to explain their results is missing. Specifically, the authors should show more than one gene in Fig. 2 to validate the "genome engineering mode" of their model and explain which parameters or terms change in their model for genome vs. plasmid RBS library construction.

Also, how is the model different for gram positive vs. gram negative bacteria? The authors neglect to explain the difference between these two models, which may provide insights into the mechanisms of translation initiation. Also, what could explain the weak correlation of the predicted and measured gene expression for low expression levels of mRFP1 in Fig. 2A? In this regime, the model predictions are not very accurate and it would be helpful if the authors could explore or discuss some potential reasons for this error.

- In regards to Fig. 3, the authors conclude that the "ribosome library calculator" can be used to identify the maximum translational capacity of a gene. RBS calculator only accounts for translation initiation and not elongation. Maximum translation capacity should be dependent on the entire transcript and not only N-terminal region of it.

However, what if the model cannot explore the high expression level regime? Without comparing their results more rigorously to a random RBS library, I do not think it is possible to distinguish whether there is an intrinsic threshold in gene expression capacity or simply the model is unable to accurately predict high expression levels. Also, could the saturation of fluorescence be a consequence of saturation of their measurement? The authors should provide data to distinguish between these possibilities. Furthermore, more than one fusion protein should be used to characterize observed the saturation effect. Surprisingly, there is no explanation for the specific choice of adhP in the text. Finally, the authors state that the "ribosome library calculator" could be used to determine if genes are codon optimized yet no data is provided to support this claim.

- Instead of plotting RBS variant number vs fluorescence plots, the authors should always plot the predicted translation initiation rate vs observed activity or fluorescence.

- The authors provide no explanation on which design specifications went into operon engineering. The RBS library was designed for single genes and cloned into an operon. One of the key questions is whether the performance of single cistron is same when cloned into an operon. Interesting question is how the RBS sequences cloned in an operon context impact the expression of downstream gene. No discussion is provided on design criteria.

As a whole the paper addresses an important topic: how to make predictive models of gene expression that allow effective exploration of complex pathway design; how to characterize the parameter space; and how to learn from that parameter space the requirements for a more focused library to improve productivity. Deeply focusing on the issues surrounding this topic would improve the paper. The large number of side points and discussions distracts from the main message and weakens the rigor of the arguments that could be made.

Reviewer #2:

In the submitted manuscript, Farasat et al develop a ribosome binding site (RBS) Library Calculator. The RBS Library Calculator uses the previously developed RBS Calculator v1.1 (Salis, *Methods Enzymol*, 2011) to create RBS libraries at a specified search resolution and that will deliver a user-defined minimum and maximum expression level for the protein of interest. The RBS Library Calculator can be used to generate libraries expressed on plasmids or libraries integrated chromosomally. Using several reporter genes, these libraries gave 76%-99% coverage of the search space over an expression level range of up to 169,000-fold. The average predicted translation initiation rates generally fall within 2.5-fold of the observed rates. The increase in coverage and dynamic range represents a significant increase over current methods. Additionally, the libraries are

effective in both *E. coli* and *B. subtilis*, at least for individual genes, suggesting that the RBS Library Calculator can be used with both gram-negative and gram-positive bacteria. The authors also applied the RBS Library Calculator to a three gene operon and once again found that the RBS Library Calculator gives better coverage of expression level space than random RBS mutagenesis does. Finally the authors looked at a three enzyme biosynthesis pathway. The RBS Library Calculator was used to vary expression levels of all three enzymes and then these libraries were assembled combinatorially. By screening 73 pathways for end product formation and then refining predictions with an additional 19 pathway variants, the authors were able to significantly increase production of the compound of interest. All data are incorporated into a biophysical model (and in some cases also a kinetic model) that can be used to determine single genotype-phenotype relationships or that can be used with metabolic flux analysis. Therefore, this manuscript should be of general interest to members of the synthetic biology and metabolic engineering communities.

A few points need to be addressed.

1. Although the author do report results using a variety of genes and *E. coli* and *B. subtilis*, a more thorough study of strain to strain and protein to protein predictions needs to be included to generate the envisioned impact (otherwise it is hard to justify several of the conclusions drawn in the current version). Specifically, fig. 1 or fig. 2 should be expanded to include multiple *E. coli* strains -- W3110, MG1655, BL21, etc. and several different proteins (RFP, GFP, lacZ, and at least one other enzyme of choice). The authors may have more creative ways to address the concern here -- which is that results will not hold up when expanded to additional strains and proteins. However, given the results presented it seems reasonable that they prediction should hold, which would suggest that the types of experimental data requested should be relatively easy to obtain.

A few minor points need to be addressed:

- 1) Several acronyms (e.g. CDS and ODE) should be written out in full before they are used.
- 2) Strain EchW2f is mentioned and is supposed to be described in Supplementary Table 3, but this strain does not appear in the table.
- 3) the authors should cite seminal work along with the more recent applications (for example in paragraph 2 of the main text as well as in section on Automated search for optimal protein expression levels...)
- 4) the third paragraph of the introduction on optimization is very poorly referenced (1 reference?). A substantial body of work exists on the issue of combinatorial explosion, in particular in the protein engineering literature.
- 5) the authors make a number of claims about an "optimal" search strategy being one that focuses on a subset of DNA sequence space that maximally spans the protein expression space. While this may be a perfectly valid assertion, the authors should add some discussion around alternative search strategies and bring in relevant theory/literature to support their discussion.
- 6) Pearson correlation coefficient is OK, but should also include a p-value to allow rapid interpretation of statistical significance of the "fit".

Reviewer #3:

This paper is a long awaited report on the important approach developed by Howard Salis et al to span protein expression space using RBS libraries in a rational manner. The method is very powerful and the paper clearly explains the motivation and the implementation. The analysis is detailed and the text and supplementary information is thorough and professional. I think this is a very appropriate contribution to MSB. My comments are aimed to help clarify some statements made by the authors that could be confusing to the readers.

- Page 3, "translation rate range on a >100,000-fold proportional scale".

It is not clear to me that such a large range is reasonable. I am aware it was stated also in Salis et al, Nat Biotech in the past. Yet, I think about it in the following way. The fastest translational rate is about 1-2 protein produced per second (due to the maximal translation rate of about 20 aa/s i.e. 60 bases/s and given that ribosomes occupy ~ 30 nucleotides in their physical footprint). So 5 orders of magnitude less is a rate of 1 protein per 10^5 seconds. That is about 1 protein per day. To what level do the author claim to be able to measure such a low value accurately? I want to make sure this is not coming from some subtraction that can result in low values as close to zero as wanted. I think even if the claim on the range will be 10 or 100 fold less it will still be very impressive and possibly more "defendable" (say >1000 or >10,000).

From sup Table 1 it seems that all FI values below 20 had associated SD>average value. This means I think that you had many negative values which are not realistic. I think it is the prudent approach to set as ~20 your minimal value and then evaluate fold change from those numbers. You will still have ~10000 fold changes which people will be more comfortable accepting which making the approach less powerful.

- Page 3, "calculates the ribosome's binding free energy to mRNAs, which is responsible for controlling its translation initiation rate."

Will be useful to give a reference and maybe a short intuitive explanation in the text why this is true. Naively if binding is stronger, translation might not start. It is probably relevant to mention that this is not expected to happen because of energy releasing stage in initiation that can overcome the energy of binding and thus the only important term is the occupancy/binding by ribosomes which is dependent on the binding dG following a Boltzmann distribution.

- Page 4 and methods section,

"by measuring reporter protein expression levels", does this mean the quantification is expression=fluorescence/OD or fluorescence per cell? The RBS strength (or translation initiation rate) seems to refer more directly to $d(\text{fluorescence})/dt/OD$. Under balanced growth (which is probably achieved reasonably well under the dilutions protocol) this will be proportional to fluorescence/OD or fluorescence per cell so I think all the results reported are valid but I think it is worth noting something of this subtlety in the text.

- Page 4,

"from low to high levels with a 49,000-fold dynamic range" here again I am curious how accurately the lower level is estimated. What are the error bars on the lower level in biological repeats measurements? The subtraction of the background in the flow cytometry seems like a delicate issue that can easily shift the fold change by an order of magnitude. Are the subtracted values and their SD given the supp. excel files?

- Page 4, "299-fold dynamic range" I think it would make more sense to write 300 given the repeatability error/uncertainty of such measurements which I guess is >0.3%.

- Page 5, Figure 1A. I think this is a wonderful figure. Minor comment, I think random search in DNA space will lead to expression levels clustered at the low levels for both enzymes so I would suggest to put the black points closer to the origin.

- Figure 1B, Where is it said what is S ? similarly P, dRBS ? Also, maybe state that the CDS is the coding sequence.

I notice that in C it mRFP while in D, E it is sfGFP. Is there somewhere a comparison of the results for same RBS set for different fluorophores to see how it varies with that context?

Where is Table 1 legend to explain what is TIR standing for?

- Page 6. "with an average error G_{total} of 1.74 kcal/mol, which is equivalent to predicting the measured translation initiation rate to within 2.2-fold". Under normal conditions (RT values), 1.7 kcal/mol is about 10 fold. I imagine here it might be different because the effective temperature might be different or the like. This should be mentioned or else the alert reader will think there is some mistake.

- Figure 3 is a strong result that I think helps explain some observations of people using RBSs in the

lab. I think this should be briefly mentioned in the abstract.

- Page 10, in future work could be interesting to estimate the rate of production in absolute units of proteins per mRNA per second but I can understand this is beyond the scope of the this paper

- P. 17, "A pathway is balanced when differential increases in enzyme expression all have the same effect on pathway productivity, which occurs when the enzymes' FCCs are equal."

I like the effort to try and define what is a balanced pathway which is a term usually used without any clear well defined meaning. Yet, I am not sure I agree with the definition given here. Having equal FCCs it means that a one percent increase in expression of any of the enzymes will have the same effect on flux. But if the enzyme levels are not the same, this translates into very different absolute expression changes. Thus an increase of one copy of one enzyme will not be the same as the increase of one copy of another enzyme. I will be happy if the authors motivate or update their definition. The alternative definition of requiring equal absolute effects will entail that the FCC will be proportional to the absolute expression levels at the "balanced state". I think this is also discussed in some MCA studies of the late Reinhardt Heinrich (but not 100% sure).

- P. 17, "an optimally balanced pathway will have nearly zero FCCs; increasing the enzymes' expression levels has a minimal impact on pathway productivity. According to the summation rule for FCCs, if control over a pathway's productivity is reduced at one step, it is correspondingly increased at another. An optimally balanced pathway has shifted control of its flux over to the upstream metabolic module controlling precursor biosynthesis."

Here again the authors do well to aim at defining the meaning of an optimal pathway. I would say that in my view the optimal pathway in this metabolic engineering context is the one that achieves the maximal productivity. In such a case the fact that further increase in enzyme levels does not translate into increase in yield might come from protein burden issues (i.e. effect on growth rate etc). Including such effects is as far as I understand, beyond the usual scope of MCA. An increase of all enzymes by a fixed factor will not increase the overall flux by the same factor because of the effect on growth rate, limited ribosomes etc. Not sure how this should be handled in this paper but thought it is important the authors will know this can be a confusing issue.

1st Revision - authors' response

02 March 2014

(see next page)

Subject: Response to Reviewers

We thank the reviewers for their careful reading of the manuscript. In response to their productive comments, we have made substantial changes throughout the manuscript to improve its focus and to provide additional data to support its conclusions. Specifically,

1. We have rewritten the introduction to focus on the challenges involved in building models for the rational design of genetic systems. The introduction is now shorter and succinctly describes the types of combinatorial explosion that have made model building extremely laborious. We have also created a new schematic (Figure 1A) to illustrate the multi-step procedure for building and using sequence-expression-activity models, using the keywords Search, Map, Design, Zoom, and Optimize to describe each step. These keywords are re-used throughout the results section to guide the reader through the process of building predictive models and using them for rational design.

2. In response to reviewers' requests, we have added a new result section and figure (*Navigation of Expression Spaces in Diverse Bacterial Species* and Figure 2) where we demonstrate the efficient searching of expression spaces within four bacterial hosts commonly used in biotechnology: *E. coli* BL21, *Pseudomonas fluorescens*, *Salmonella Typhimurium* LT2, and *Corynebacterium glutamicum*. We find that the biophysical model accurately predicts translation initiation rates in these diverse hosts (average error $\Delta\Delta G_{\text{total}}$ of 1.61 kcal/mol, Pearson R^2 is 0.89) though the ribosome's interactions with mRNA differ between the gram-positive *C. glutamicum* and the other gram-negatives. This result fits cleanly with our previous results demonstrating the efficient search of sequence-expression spaces in genetic systems with multiple proteins, encoded in plasmids and genomes. Altogether, we have characterized 646 genetic system variants to illustrate how our efficient search approach circumvents several types of combinatorial explosion.

3. The reviewers noted that the results shown in Figure 3 of the previous manuscript seemed out of place. We agree that proposing a new way to measure the maximum translation rate capacity of a codon-optimized gene diverges from the intended focus of the manuscript, and that additional data is needed to firmly support the proposed mechanism for the observed plateau in protein expression. We have removed this section and its figure from the results. We plan to address this question in another manuscript with additional data. In the revised manuscript, Figure 3 now demonstrates the navigation of sequence-expression spaces by modifying the genomes of gram-positive (*Bacillus subtilis*) and gram-negative (*E. coli*) bacteria.

4. We have added some explanatory labels to the graphs shown in Figure 4 to coincide with the workflow detailed in Figure 1A. We believe this will help the reader understand the entire process for building and using models to access targeted behaviors, both within the characterized sequence-expression-activity space (interpolation) and outside of it (extrapolation).

5. The reviewers noted that optimization of the metabolic pathway, using the predictive model, was impressive and potentially useful (Figure 4). To demonstrate additional ways that the predictive model can be used for rational design, we have added two new results, accompanied by Figures 5 to 6.

5a. First, we show that the predictive sequence-expression-activity model can predict the effects of using a different promoter (IPTG-inducible PlacO1) to drive the expression of four different pathway variants, encoded in bacterial operons. Consistent with the model predictions, the relationship between transcription rate and pathway activity can be non-linear, and depends on the pathway variants' *crtEBI* translation rates. These results are particularly important as transcriptional regulation is often used to dynamically control pathway expression levels in response to changing environmental conditions. By using the model, one can rationally select a pathway variant whose transcription rate vs. activity response behavior is desirable.

5b. Second, to demonstrate our approach's efficiency & scalability and to further illustrate the applications of the flux control coefficient calculations, we applied efficient search and the RBS Library Calculator to systematically varying the expression of the genome-encoded enzyme *dxs*, which catalyzes the first committed step in isoprenoid biosynthesis, while expressing either an optimally balanced *crtEBI* pathway or an imbalanced *crtEBI* pathway, as defined by their FCCs. The results clearly show that increases in isoprenoid biosynthesis rates will greatly increase neurosporene productivity, but only when the downstream pathway is optimally balanced. These results are extremely important to understanding how upstream and downstream metabolic pathways interact, and to demonstrate a quantitative approach for selecting pathway variants with the optimal behaviors.

6. Finally, we thought it was intriguing that human-directed mutagenesis and natural evolution both navigate the same sequence-expression-activity space. In the discussion section, we illustrate how evolution could shape the pathway's productivity by using the model to calculate the effects of all possible single, double, and triple mutations to the *crtEBI* ribosome binding site sequences (each 35 nt long), starting from the optimally balanced pathway variant. Overall, mutations are expected to decrease the pathway's productivity, consistent with its globally optimal location in expression-activity space. However, these results do suggest an optimization approach for designing sequences to be robust to evolutionary change, which is an excellent topic for another paper.

Altogether, these changes have focused the manuscript towards describing a new approach for building predictive sequence-expression-activity models, and demonstrating how they are used to rationally design and optimize multi-protein genetic systems. In the following pages, we respond to each reviewer comment, and specifically state what modifications were made to the text in response. **Below, modifications are highlighted in bold blue.**

Reviewer #1:

The main idea to use a computational model based on biophysical interactions to guide experimental design is interesting and important. Also, developing methods for rationally sampling the design space are needed that can yield more information content from minimal samples. As a whole the paper

addresses an important topic: how to make predictive models of gene expression that allow effective exploration of complex pathway design; how to characterize the parameter space; and how to learn from that parameter space the requirements for a more focused library to improve productivity. Deeply focusing on the issues surrounding this topic would improve the paper. The large number of side points and discussions distracts from the main message and weakens the rigor of the arguments that could be made.

As described above, we have re-organized the text and added new results to focus the manuscript's conclusions on the efficient building of sequence-expression-activity models, and their use in rational design and optimization. We have removed indirectly related results, for example, measuring the maximum translation rate capacity, though these results are important to achieving full control over enzyme expression levels for rational design and will be the topic of another paper.

The author's conclusions based on Fig. 2 and 3 are not supported by sufficient data and controls. In addition, important information to explain their results is missing. Specifically, the authors should show more than one gene in Fig. 2 to validate the "genome engineering mode" of their model and explain which parameters or terms change in their model for genome vs. plasmid RBS library construction.

The RBS Library Calculator's Genome Editing mode has been validated using two reporter genes: the fluorescent *mRFP1*, and the enzymatic *lacZ*. The biophysical model is unchanged when optimizing RBS libraries for plasmids vs. genomes. As explained in the text, the Genome Editing mode uses a set of optimization constraints to design optimal RBS libraries with adjacent/nearby degenerate mutations. Altogether, the RBS Library Calculator was validated using four different reporters (*mRFP1*, *sfGFP*, *cfp*, *lacZ*). Further, the RBS Library Calculator was employed to systematically vary the expression of four enzymes (*crtE*, *crtB*, *crtI*, and *dxs*).

Also, how is the model different for gram positive vs. gram negative bacteria?

Here, the biophysical model of translation is the same for both gram-positive and gram-negative bacteria. The only difference is the 3' 16S rRNA sequence (last 9 nt) that is inputted into the model. The 3' 16S rRNA sequence is ACCTCCTTA for *E. coli* and *P. fluorescens*; and it is ACCTCCTTT for *C. glutamicum* and *B. subtilis*.

Also, what could explain the weak correlation of the predicted and measured gene expression for low expression levels of mRFP1 in Fig. 2A? In this regime, the model predictions are not very accurate and it would be helpful if the authors could explore or discuss some potential reasons for this error.

Quantitatively, the predictive accuracy of the biophysical model is similar at the low end of expression vs. the high end of expression. The biophysical model is not perfectly accurate, and its error is explicitly stated in the text. The reviewer should note that low expression levels are difficult to measure, and accordingly, the measurements at the low end of the expression space have higher error bars. As a result, when comparing measurements to predictions using a correlation metric, the metric must also

account for the variation in observable due to measurement precision. **To address this concern, we have added two-tailed significance tests in the figure legends together with Pearson R^2 calculations.**

Instead of plotting RBS variant number vs fluorescence plots, the authors should always plot the predicted translation initiation rate vs observed activity or fluorescence.

In Figures 2, 3, and 6, we plot predicted translation initiation rates vs. observed protein fluorescence levels. However, in Figure 1, the key point is that the RBS Library Calculator designs degenerate sequences with the smallest number of RBS variants to navigate the widest expression space. We compare high, medium, and low resolution searches using the same optimization algorithm with different input parameters. Thus, the number of RBS variants is central to the point, and is shown on the x-axis. The comparison to model predictions are also shown, labeled as diamonds. The errors in model predictions are explicitly described in the text and figure legend.

The authors provide no explanation on which design specifications went into operon engineering. The RBS library was designed for single genes and cloned into an operon. One of the key questions is whether the performance of single cistron is same when cloned into an operon. Interesting question is how the RBS sequences cloned in an operon context impact the expression of downstream gene. No discussion is provided on design criteria.

In the original manuscript, a paragraph in the Discussion discussed design criteria, stated that "Potentially confounding interactions that affect protein expression are minimized by eliminating long single-stranded RNA regions or long RNA duplexes that may reduce mRNA stability, by ensuring that translation elongation is not rate-limiting, and by ensuring that mRNAs are always translated to protect them from RNase activity. By incorporating these design rules into the engineering of bacterial operons, one can achieve proportional control of protein expression by manipulating only RBS sequences. As additional biophysical rules continue to be developed (Espah Borujeni et al), they are incorporated into the forward design process, and can improve the accuracy of predictions on previously designed sequences. Thus, computational design can evolve concomitantly with our understanding of gene expression and the development of new DNA assembly, genome mutagenesis, and genome synthesis techniques to accelerate the engineering of large genetic systems."

We think Fig. 4 is the notable contribution of this paper and the motivation is similar to the recent Lee, ME et al. NAR 2013 paper that engineered the violacein pathway using a statistical model. Since this seem the main contribution it would seem that some detailed comparison of the present results with the approaches taken by the Lee paper in yeast are in order.

There are two key differences between our approach and the one described in Lee et. al.

First, we use a biophysical model coupled to optimization to perform de novo design of optimized RBS libraries. We show the ability to design many RBS libraries that controlled expression with a 1000- to >10,000-fold ranges, depending on the input parameters. If more RBS libraries are needed, more are designed; particularly, the targeted translation rates can be adjusted as needed, including very high

translation rates. For example, the RBS libraries controlling *crtI* and *dxs* expression have maximum translation rates that exceed 100,000 au. In Lee et. al., five previously characterized yeast promoters are re-used to control the expression of a 5-enzyme violacein biosynthesis pathway. The five promoters have transcription rates that span an approximate 1000-fold range. The same five yeast promoters are used to drive the expression of the five enzymes in a combinatorial fashion, generating up to 3125 pathway variants. We note that additional promoters were characterized, but the yeast promoter with the highest transcription rate was selected to be one of the five. To access higher expression levels, additional yeast promoters will need to be designed and characterized.

Second, we use chemical kinetics to model the enzymes' reaction rates and to predict the relationship between enzyme expression levels and pathway flux. Lee et. al. uses linear regression to find correlations between enzyme expression levels and product accumulation. Our approach develops a mechanistic model, whereas the Lee et. al. approach develops a statistical/informatic model. Other types of non-mechanistic models are also available, including computational geometry models. There are substantial differences between all three approaches, including the ability to design genetic systems with intermediate activities (interpolation) or with higher activities (extrapolation). **We have added a few paragraphs in the Discussion section to quantitatively compare these differences, using the same 73 *crtEBI* pathway data-set as the training set.** In particular, we find that non-mechanistic models can not perform extrapolative predictions without assuming that the expression-activity relationship is linear and independent. We also found that log-transforming productivity/activity measurements substantially alters the structure of a training data-set. While a statistical model could predict the log-transformed pathway productivity data-set with an error of only 10%, the same modeling approach would fit randomly generated log-transformed pathway productivity data with a similar error of 15%. Thus, the ability of any model to fit a training data-set needs to be compared to the same training process on a randomly generated data-set in order to quantify the comparison.

We think the regression approach in Lee is likely more efficient than the present one when the landscape is additive (non-epistatic) but the RBS library calculator approach may be a better method for optimizing systems with nonlinear production landscapes. The violacein pathway seems to be fairly linear in its response to expression (all main pathway components high seems to be pretty good). It would have been nice to see a more complete analysis of the neurosporene pathway parameter space. The utility of these approaches is deeply tied, we think, to the properties of these parameter spaces that map expression element activity to productivity for particular pathways.

According to the physical theory of chemical kinetics, all metabolic pathways have non-linear relationships between enzyme expression level and pathway flux. Under restricted conditions, it may appear that changes in enzyme expression give proportional (linear) changes in pathway flux (e.g. high flux control coefficients). Importantly, the conditions that we care about are the ones where all enzymes are no longer rate-limiting the pathway's flux (low flux control coefficients). Under these conditions, the relationship between enzyme expression and pathway flux is highly non-linear because any enzyme could become rate-limiting with a small change in expression level. The example of the violacein biosynthesis pathway shows that the optimal enzyme expression levels have not been reached because

further increases in enzyme expression always increased pathway productivity, including the promoter with the highest transcription rate. At a critical point where the flux control coefficients are zero, the violacein productivity should reach a plateau where additional increases will require metabolic engineering of the tryptophan biosynthesis pathway.

Even if the pathway does have a complex topology to the parameter space, we suspect the proposed method by Farasat, I et al. does not scale well with the dimension of the pathway. For higher dimension systems, small modifications to the Lee method may be more useful. Again-- the interest here would be in the dissection of this issue.

Using the elementary mass action kinetics formalism, the addition of a fully reversible enzyme-catalyzed chemical reaction adds four kinetic parameters to the model. However, two of these kinetic parameters are eliminated via model reduction, leaving two independent parameters. Therefore, for each additional reaction in the pathway, the number of unknown parameters increases by two. This is a linear relationship that scales well. **We have added sentences in the text describing the number of unknown parameters when modeling metabolic pathways or genetic circuits.** In contrast, the number of unknown coefficients in the category-based linear regression model is equal to the number of unique genetic parts used to control expression multiplied by the number of proteins. In Lee et. al., the same five promoters are re-used to control the expression of the five enzymes, yielding twenty-five coefficients. However, to avoid homologous recombination, it would be more desirable to use non-repetitive genetic parts to control the expression of different enzymes. Adding more genetic parts to the training set will necessarily increase the number of unknown coefficients in the linear regression model. Adding an additional enzyme also increases the number of unknown coefficients. The scaling is quadratic, not linear.

Further, model reduction and model identification of differential equation-based models is a well studied engineering topic. There are several techniques for ensuring that a multi-dimensional model is an accurate representation of the steady-state, dynamical, and/or stochastic dynamical behavior of a real system without over- or under- fitting. We used some of these techniques in our study, resulting in repeated convergence to a unique parameter solution.

Some of the figures of the present paper are very confusing and unclear (especially Fig. 1A and Fig. 4) with some extraneous details that are only tangentially related to the point (e.g. the reference to the game of "Battleship" in Fig. 1 may seem clever but it detracts from the point). In addition, overall we found that the writing is not very clear and could be improved (for example, explaining the relationship between the kinetic model and the RBS library calculator and providing sufficient detail to understand the figures).

We have introduced a schematic in Figure 1A to explain the overall process for building and using sequence-expression-activity models. We have removed the Battleship figures as tangential to the main point of the manuscript. We have also added explanatory labels to Figure 4 to coincide with Figure 1A.

In regards to Fig. 3, the authors conclude that the "ribosome library calculator" can be used to identify the maximum translational capacity of a gene. RBS calculator only accounts for translation initiation and not elongation. Maximum translation capacity should be dependent on the entire transcript and not only N-terminal region of it. However, what if the model cannot explore the high expression level regime? Without comparing their results more rigorously to a random RBS library, I do not think it is possible to distinguish whether there is an intrinsic threshold in gene expression capacity or simply the model is unable to accurately predict high expression levels. Also, could the saturation of fluorescence be a consequence of saturation of their measurement? The authors should provide data to distinguish between these possibilities. Furthermore, more than one fusion protein should be used to characterize observed the saturation effect. Surprisingly, there is no explanation for the specific choice of adhP in the text. Finally, the authors state that the "ribosome library calculator" could be used to determine if genes are codon optimized yet no data is provided to support this claim.

As shown in the new Figures 1, 2, and 3A, the biophysical model of translation is similarly accurate at high translation initiation rates, compared to low or medium translation initiation rates. There is a mechanistic reason for observed plateaus in protein expression that is independent of increasing translation initiation. The data to test the proposed mechanism would significantly add to the manuscript's length. **This result section and the old Figure 3 have been removed. This topic and these questions will be addressed in another manuscript.**

Reviewer #2:

*Although the author do report results using a variety of genes and *E. coli* and *B. subtilis*, a more thorough study of strain to strain and protein to protein predictions needs to be included to generate the envisioned impact (otherwise it is hard to justify several of the conclusions drawn in the current version). Specifically, fig. 1 or fig. 2 should be expanded to include multiple *E. coli* strains -- W3110, MG1655, BL21, etc. and several different proteins (RFP, GFP, lacZ, and at least one other enzyme of choice). The authors may have more creative ways to address the concern here -- which is that results will not hold up when expanded to additional strains and proteins. However, given the results presented it seems reasonable that they prediction should hold, which would suggest that the types of experimental data requested should be relatively easy to obtain.*

Following the reviewer's request, we have characterized the strain-to-strain differences in the hosts *E. coli* BL21, *Pseudomonas fluorescens*, *Salmonella Typhimurium* LT2, and *Corynebacterium glutamicum* by measuring *mRFP1* expression from synthetic cassettes on broad-host vectors. When inputting the appropriate 3' 16S rRNA into the biophysical model, we find that relative translation initiation rates are well-predicted with a similar accuracy as in *E. coli* DH10B and MG1655 (Figure 2). We observe that the absolute *mRFP1* protein fluorescences for all RBS variants can shift upwards or downwards based on host-dependent promoter differences, vector copy number differences, or differences in host growth rates. Altogether, the biophysical model predictions were validated using four different reporters (*mRFP1*, *sfGFP*, *cfp*, *lacZ*). Further, the biophysical model was employed to systematically vary the expression of four enzymes (*crtE*, *crtB*, *crtI*, and *dxs*).

Several acronyms (e.g. CDS and ODE) should be written out in full before they are used.

We have written out the names protein coding sequence and ordinary differential equation where they are first used.

Strain EchW2f is mentioned and is supposed to be described in Supplementary Table 3, but this strain does not appear in the table.

The strains EcNR2 and EchW2f have been added to Supplementary Table 2, which now contains the list of Strains and Plasmids.

The authors should cite seminal work along with the more recent applications (for example in paragraph 2 of the main text as well as in section on Automated search for optimal protein expression levels...).

We have added citations to several seminal works from which we draw upon, particularly in the systems biology and metabolic modeling fields. The vision and specific results from Profs. Kitano, Bailey, Strohmman, Fell, and Westerhoff are fundamental to our study.

The third paragraph of the introduction on optimization is very poorly referenced (1 reference?). A substantial body of work exists on the issue of combinatorial explosion, in particular in the protein engineering literature.

We have referenced works by Profs. Bailey and Stephanopoulos that show how combinatorial explosions affects our ability to understand and engineer changes to metabolism. Combinatorial explosion also stymies protein engineering efforts, though biocatalysis design and its challenges are not addressed in our study.

The authors make a number of claims about an "optimal" search strategy being one that focuses on a subset of DNA sequence space that maximally spans the protein expression space. While this may be a perfectly valid assertion, the authors should add some discussion around alternative search strategies and bring in relevant theory/literature to support their discussion.

In the main text's section "Efficient search in multi-dimensional expression spaces", we compare different approaches to designing degenerate sequences for searching expression spaces. Our comparisons focus on recently used strategies for changing protein expression levels when introducing a library of sequences, and utilize Monte Carlo calculations to quantitatively compare search coverages. **In the Discussion section, we have also added a comparison between the use of different types of system-level models to find the optimal enzyme expression levels of pathways.**

Pearson correlation coefficient is OK, but should also include a p-value to allow rapid interpretation of statistical significance of the "fit".

Two-tailed significance tests have been performed alongside all Pearson correlation coefficient calculations and added to the figure legends. These statistical tests reject the null hypothesis (p values are very small in all cases as stated in the text) and confirm that the observed linear relationship

between predicted translation initiation rates and measured protein expression levels does not occur by chance.

Reviewer #3:

This paper is a long awaited report on the important approach developed by Howard Salis et al to span protein expression space using RBS libraries in a rational manner. The method is very powerful and the paper clearly explains the motivation and the implementation. The analysis is detailed and the text and supplementary information is thorough and professional. I think this is a very appropriate contribution to MSB. My comments are aimed to help clarify some statements made by the authors that could be confusing to the readers.

Page 3, "translation rate range on a >100,000-fold proportional scale". It is not clear to me that such a large range is reasonable. I am aware it was stated also in Salis et al, Nat Biotech in the past. Yet, I think about it in the following way. The fastest translational rate is about 1-2 protein produced per second (due to the maximal translation rate of about 20 aa/s i.e. 60 bases/s and given that ribosomes occupy ~ 30 nucleotides in their physical footprint). So 5 orders of magnitude less is a rate of 1 protein per 10^5 seconds. That is about 1 protein per day. To what level do the author claim to be able to measure such a low value accurately? I want to make sure this is not coming from some subtraction that can result in low values as close to zero as wanted. I think even if the claim on the range will be 10 or 100 fold less it will still be very impressive and possibly more "defendable" (say >1000 or >10,000).

We agree with your calculations, but would like to add that it should be 1-2 proteins per second per mRNA transcript. At very low translation initiation rates, we need to express a reporter protein on a high-copy plasmid with a suitably "strong" promoter in order to observe the reporter's activity and distinguish it from the background. Even so, we never treat protein fluorescences below 1 au as distinguishable from background fluorescence, although they are reproducibly low. **Regardless, we have modified the main claim to "efficiently searching a multi-protein expression space across a >10,000-fold range" when the optimization algorithm is instructed to do so (e.g. when the minimum and maximum target translation rates are at least 10,000-fold apart).**

From sup Table 1 it seems that all FI values below 20 had associated SD>average value. This means I think that you had many negative values which are not realistic. I think it is the prudent approach to set as ~20 your minimal value and then evaluate fold change from those numbers. You will still have ~10000 fold changes which people will be more comfortable accepting which making the approach less powerful.

Distinguishing low protein fluorescences from background autofluorescence is a common challenge. **To avoid over-stepping the data, we have altered the overall claim to controlling translation initiation rates across a >10,000-fold scale.** We should also note that 100,000 is not the maximum possible translation initiation rate. Here, we have increased *crtI* and *dxs* translation initiation rates above 100,000.

Page 3, "calculates the ribosome's binding free energy to mRNAs, which is responsible for controlling its translation initiation rate." Will be useful to give a reference and maybe a short intuitive explanation in the text why this is true. Naively if binding is stronger, translation might not start. It is probably relevant to mention that this is not expected to happen because of energy releasing stage in initiation that can overcome the energy of binding and thus the only important term is the occupancy/binding by ribosomes which is dependent on the binding dG following a Boltzmann distribution.

We would also like to note that recruitment of the 50S large subunit and GTP hydrolysis introduces an external source of available free energy that prevents the 30S small subunit from being trapped on the mRNA as it ratchets forward during translation elongation. **We have referenced our recent paper "Translation rate is controlled by coupled trade-offs between site accessibility, selective RNA unfolding and sliding at upstream standby sites" that explains some of these mechanisms.**

Page 4 and methods section, "by measuring reporter protein expression levels", does this mean the quantification is $\text{expression} = \text{fluorescence}/\text{OD}$ or fluorescence per cell? The RBS strength (or translation initiation rate) seems to refer more directly to $d(\text{fluorescence})/dt/\text{OD}$. Under balanced growth (which is probably achieved reasonably well under the dilutions protocol) this will be proportional to $\text{fluorescence}/\text{OD}$ or fluorescence per cell so I think all the results reported are valid but I think it is worth noting something of this subtlety in the text.

Yes, the measurement is fluorescence per cell. Under steady-state growth conditions used here, protein expression levels are proportional to translation initiation rates. To relate translation initiation rate to the net production rate of protein fluorescence ($d\text{fluorescence}/dt/\text{OD}$) would also require taking into account dilution by growth rate. $d\text{fluorescence}/dt/\text{OD}$ eventually approaches zero as steady-state is reached. We have modified the text to include the steady-state growth condition.

Page 4, "from low to high levels with a 49,000-fold dynamic range" here again I am curious how accurately the lower level is estimated. What are the error bars on the lower level in biological repeats measurements? The subtraction of the background in the flow cytometry seems like a delicate issue that can easily shift the fold change by an order of magnitude. Are the subtracted values and their SD given the supp. excel files?

In our measurements, the background autofluorescence is repeatedly measured and is reproducible. We subtract the average autofluorescence from the average fluorescence to obtain average protein fluorescences. If the average protein fluorescence is below 1 au, we do not divide by this small number. Instead, we divide by 1 au. This prevents the concern the reviewer has noted, where somewhat arbitrary autofluorescence values could be subtracted to achieve very low protein fluorescences and very high dynamic ranges. We do not do this. Besides the dishonesty, it's fairly pointless and does not contribute to our overall goal of rationally engineering genetic systems. I can't speak for others

- Page 4, "299-fold dynamic range" I think it would make more sense to write 300 given the repeatability error/uncertainty of such measurements which I guess is $>0.3\%$.

The sentence has been changed from "yielding protein expression levels from 63 to 49,000 au (299-fold dynamic range)" to "yielding protein expression levels from 63 to 49,000 au (778-fold dynamic range)". There was an error in the previous calculation.

Page 5, Figure 1A. I think this is a wonderful figure. Minor comment, I think random search in DNA space will lead to expression levels clustered at the low levels for both enzymes so I would suggest to put the black points closer to the origin. Figure 1B, Where is it said what is S ? similarly P, dRBS ? Also, maybe state that the CDS is the coding sequence. I notice that in C it mRFP while in D, E it is sfGFP. Is there somewhere a comparison of the results for same RBS set for different fluorophores to see how it varies with that context? Where is Table 1 legend to explain what is TIR standing for?

While we also liked the previous Figure 1A, it was perhaps too abstract and conceptual. We believe the new Figure 1A provides a concrete description for how building and using sequence-expression-activity models enables rational design and optimization. **We have modified the legend of Figure 1 to include the acronyms dRBS and CDS alongside their definitions. A single new sentence describes how the biophysical model relates mRNA sequence to the ribosome's Gibbs binding free energy, and to protein expression level Table 1 has been removed to reduce the manuscript's length.**

Page 6. "with an average error $\delta\delta G_{total}$ of 1.74 kcal/mol, which is equivalent to predicting the measured translation initiation rate to within 2.2-fold". Under normal conditions (RT values), 1.7 kcal/mol is about 10 fold. I imagine here it might be different because the effective temperature might be different or the like. This should be mentioned or else the alert reader will think there is some mistake.

Regarding the predictions tested in Figure 1, we note in the text that the apparent beta is 0.45 mol/kcal. In Figure 2, we also find that the apparent beta is very similar across different bacterial hosts (0.42 +/- 0.02).

Figure 3 is a strong result that I think helps explain some observations of people using RBSs in the lab. I think this should be briefly mentioned in the abstract.

While we agree with the reviewer, we feel that there are many competing hypotheses regarding "codon optimization". We believe that a new way to quantify what it means to be "codon optimized" should be described in a separate article with additional data to support the conclusions.

Page 10, in future work could be interesting to estimate the rate of production in absolute units of proteins per mRNA per second but I can understand this is beyond the scope of the this paper.

We agree.

P. 17, "A pathway is balanced when differential increases in enzyme expression all have the same effect on pathway productivity, which occurs when the enzymes' FCCs are equal." I like the effort to try and define what is a balanced pathway which is a term usually used without any clear well defined meaning. Yet, I am not sure I agree with the definition given here. Having equal FCCs it

means that a one percent increase in expression of any of the enzymes will have the same effect on flux. But if the enzyme levels are not the same, this translates into very different absolute expression changes. Thus an increase of one copy of one enzyme will not be the same as the increase of one copy of another enzyme. I will be happy if the authors motivate or update their definition. The alternative definition of requiring equal absolute effects will entail that the FCC will be proportional to the absolute expression levels at the "balanced state". I think this is also discussed in some MCA studies of the late Reinhardt Heinrich (but not 100% sure).

We have modified the text describing our definition of the flux control coefficients. As stated, "FCCs quantify how differential fold-changes in enzyme expression control a pathway's overall productivity, and vary depending on the enzymes' expression levels" that fits the corresponding mathematical definition of $d\log P/d\log E$, equivalent to $dP/dE * (E/P)$. As noted by the reviewer, this definition accounts for intrinsic differences in enzyme activities that can lead to equally expressed enzymes catalyzing reactions at different rates.

P. 17, "an optimally balanced pathway will have nearly zero FCCs; increasing the enzymes' expression levels has a minimal impact on pathway productivity. According to the summation rule for FCCs, if control over a pathway's productivity is reduced at one step, it is correspondingly increased at another. An optimally balanced pathway has shifted control of its flux over to the upstream metabolic module controlling precursor biosynthesis." Here again the authors do well to aim at defining the meaning of an optimal pathway. I would say that in my view the optimal pathway in this metabolic engineering context is the one that achieves the maximal productivity. In such a case the fact that further increase in enzyme levels does not translate into increase in yield might come from protein burden issues (i.e. effect on growth rate etc). Including such effects is as far as I understand, beyond the usual scope of MCA. An increase of all enzymes by a fixed factor will not increase the overall flux by the same factor because of the effect on growth rate, limited ribosomes etc. Not sure how this should be handled in this paper but thought it is important the authors will know this can be a confusing issue.

In our definition of the FCCs, we have replaced reaction rates with pathway productivity. It is true that this expands MCA beyond examining the effects of changing enzyme levels on individual reaction rates towards investigating their effects on overall pathway productivity. Interestingly, even without including growth rate changes or competition for shared resources, the model calculates that excess enzyme expression can sequester intermediate metabolites as substrate-enzyme complex, which will cause a decrease in pathway productivity and correspondingly negative FCCs. The data shown in the newly introduced Figure 5 is consistent with these model calculations. At the very least, using FCCs allows the Metabolic Engineering community to better quantify how enzyme expression levels will affect pathway productivity, including when "less is more".

Sincerely,

Howard Salis, Assistant Professor of Biological Engineering and Chemical Engineering

Thank you again for submitting your work to Molecular Systems Biology. We have now heard back from the three referees who agreed to evaluate your manuscript. As you will see from the reports below, while the main concerns of reviewers #2 and #3 have been satisfactorily addressed, reviewer #1 lists a number of issues that need to be dealt with.

These issues regard:

- Discussing the biophysical rules and/or design specifications in the context of operon engineering. To avoid a lengthy discussion on this topic in the main text, we would suggest adding to the discussion a 'box item' including a few bullet points and/or illustrations.

- Moving the description of the results depicted in Fig. 7 from the "Discussion" to the "Results" section.

- Reviewer #1 lists a number of "other points" that refer to providing clarifications and/or including modifications (i.e. modifying some of the figures and figure legends to increase clarity, rephrasing a few sentences).

Concerning the comment of referee #1 about moving the results from different bacteria (Fig. 2B) into the supplementary information, we think that this is not necessary.

REFEREE REPORTS

Reviewer #1:

"Efficient search, mapping, and optimization of multi-protein genetic systems in diverse bacteria" (manuscript number MSB-13-4955R).

The authors have addressed most comments raised by reviewers.

- Specifically, the authors have performed additional experiments as requested by other reviewers. Authors have designed RBS library controlling mRFP1 expression and characterized them in *E. coli* BL21, *P. fluorescens*, *S. typhimurium* LT2, and *C. glutamicum*. It is impressive and not a little surprising that the RBS calculator predicts the mRFP expression so accurately in these gram-positive and gram-negative organisms by simply changing the input 3' sequence of 16S RNA.

- The authors have also performed 2 additional sequence-activity models: computational geometry (Voronoi polygons) and statistical linear regression model. In supplementary material they detail these models, compare their performances with the combined (biophysical and kinetic) modeling approach they have used. The authors argue that the kinetic model performs much better, especially in terms of interpolation or extrapolation prediction power.

MAIN POINT:

The summary is that the paper has enough strong results to merit publication in MSB. However, there are a number of issues that make the delivery of the main points of the paper difficult to extract and evaluate. While we don't want to force another round of major revision we hope the authors will consider the points below very carefully.

There are still too many weak results in this paper that are not sufficiently supported by data therefore, ideally, the paper should be broken into multiple manuscripts OR many figures should be moved to the supplementary material (e.g. showing that the RBS calculator works in other bacteria

should be its own story with detailed explanation of the differences in the model between gram positive and gram negative bacteria OR put in the supplementary material because it is tangential to the point, detailed characterization of how RBS sequences affect genes in operons - see below, using RBS library calculator to optimize metabolic pathways). We think the third major point is the strongest and the paper should focus on this topic.

We are worried that a number of claims are made in a strong voice that should not be made without rigorous data to support these statements. For example, the authors claim that they can use SEAMAP to "guide the selection of a regulated promoter to dynamically control a pathway's." (this sentence is also grammatically incorrect). Where is the data to support this claim?

Nonetheless, the core of the manuscript is quite interesting and important. The authors design and predict RBS parts performance (translation initiation rate), use them to build synthetic operon and predict the output of metabolic pathway. Overall, the designability and predictability of synthetic operon are the most important aspects of this work.

The authors design total 4 operons in this work (cfp-mrfp1-gfpmut3b operon using randomized RBSs and using RBS library outputs; and crtE-crtB-crtI operon using search and zoom modes). In each case, authors used RBS library calculator in 'search'/'zoom' mode to generate RBS sequences that yield diverse expression of individual genes, and then assembled them into thousands of pathway variants (operons) before characterizing their performance. Here a number of interesting questions arise: how does each of these RBS sequences retain their rank orders in synthetic operon context? What is the rank and absolute error in each RBS's performance? How does quantitative comparison between performance of each RBS sequence with respect to 3 different genes between the RBS library calculator output and random library looks like? Etc. One might expect this rigorous testing of calculations from the claims otherwise made in the paper.

There are many published studies that have attempted, with poor success, to design a predictably operating synthetic operon with genes coding for metabolic pathway enzymes. We had asked in earlier version of manuscript, a series of questions about how various design specifications were considered in building an operon; how performance of single cistron changed when cloned into an polycistronic operon; how does gene position impact operon or pathway performance and how RBS sequences designed for one gene (and cloned in an operon context) impact the expression of other genes of operon? These are important points for understanding the operation of a synthetic operon, the interplay between transcription and translation processes. Ideally, these lessons will help community in improvising future operon designs. This work fails to address any design specifications needed to build predictably operating metabolic pathway and it appears that everything worked as designed perfectly in the first attempt.

To our earlier comment on this topic, the authors give a fairly vague and non-responsive argument, which is also missing from this version (our question in italics and the author's reply is blue italics):

The authors provide no explanation on which design specifications went into operon engineering. The RBS library was designed for single genes and cloned into an operon. One of the key questions is whether the performance of single cistron is same when cloned into an operon. Interesting question is how the RBS sequences cloned in an operon context impact the expression of downstream gene. No discussion is provided on design criteria.

In the original manuscript, a paragraph in the Discussion discussed design criteria, stated that

"Potentially confounding interactions that affect protein expression are minimized by eliminating long single-stranded RNA regions or long RNA duplexes that may reduce mRNA stability, by ensuring that translation elongation is not rate-limiting, and by ensuring that mRNAs are always translated to protect them from RNase activity. By incorporating these design rules into the engineering of bacterial operons, one can achieve proportional control of protein expression by manipulating only RBS sequences. As additional biophysical rules continue to be developed (Espah Borujeni et al), they are incorporated into the forward design process, and can improve the accuracy of predictions on previously designed sequences. Thus, computational design can evolve concomitantly with our understanding of gene expression and the development of new DNA assembly, genome mutagenesis, and genome synthesis techniques to accelerate the engineering of large genetic systems."

We would like to know were these biophysical rules applied or not. From the current 'main text', 'supplementary material' and 'methods' section it doesn't appear that these rules have been applied.

If not- how did they prove their multicistronic "designs" operated as predicted? In that case, lessons learned (if any) in this work cannot be extended to any different pathway. If that is true then, it appears like by using cleverly optimized RBS library, authors screened for desired outputs. This is no different than current practices that screen for desired activity (for example: PMID: 16845378).

If yes- then how did they do it? How did they classify which RNA duplexes reduce mRNA stability and which ones improve the transcript stability? How does stronger RBS1-gene1 impact expression of downstream weak RBS2-gene2 ? or how does weaker RBS1-gene1 impacts expression of downstream stronger RBS2-gene2 and terminal medium RBS3-gene3?

If the authors did use SOME method to "eliminate long single-stranded RNA regions" between intergenic regions, how did they modulate/regulate the low ribosome traffic in the case of weaker RBS1-gene1 design, which will have naked mRNA and may lead to faster degradation of downstream transcript. This may also impact termination of transcription process as both transcription and translation are coupled in E coli.

How did the authors made sure that "translation elongation is not rate-limiting"? As far as the published manuscript and this current version of RBS calculator doesn't model or include parameters for translation elongation process! Do we really know any robust rules that will make translation elongation 'not limiting' for diverse set of genes?

We really do not understand how the authors "ensured that mRNAs are always translated to protect them from RNase activity" especially, when the strength of RBS1 and RBS2 are weaker than RBS3; and other combinations of such. How do the authors take into account 'rho-dependent transcription termination activity' when ribosome traffic is minimal for weaker upstream translation events? Do we really know of any proven design rules that can make any mRNA always translated insulated from neighboring context?

OTHER POINTS:

- No line numbers and page numbers. So it is challenging to review the manuscript (second time!).
- Results are presented in the discussion section
- The authors use the word 'optimized RBS library' and do not really explain what do they mean by 'optimized'?
- It seems unlikely that "...these observations suggest that the free energies of in vivo RNA-RNA interactions remain the same regardless of the host organism, including the effects of molecular crowding on binding events." There are many papers showing that host context changes the behavior of synthetic pathways. This conclusion seems contrary to the findings of these studies. Again, there should be a paper dedicated to exploring this topic in detail.
- Figure 1A is very confusing and needs to be changed to clearly show the work flow of this method
- Figure 2 legend has repeat (B) section.
- Figure 4 legend A. where is the 'search' mode (left) and 'zoom' mode (right)?
- Throughout manuscript authors have used varied predicted translation rates and do not clarify the decision to do so. For example, under 'Mapping the sequence-expression-activity space of a multi-enzyme pathway' section, ' optimized RBS libraries were designed to vary CrtE, B and I from 445 to

72000 au, 3 to 20,000 au and 97 to 203000 au'. Here, why and how are these lower values chosen?

- Odd sentence above Figure 6 (since line numbers are not given in the draft): "According to the model, the pathway variants' productivities will increase with promoter induction (Figure 5C, left), though variants with optimal translation rates will have higher productivities, and at lower transcription rates"
- Did the authors measure any enzyme levels? Sentence above Figure 6 (since line numbers are not given in the draft): "Pathway variants with optimal translation rates had higher productivities, and achieved maximum activity at a lower transcription rate. However, additional increases in transcription lowered their productivities, due to excess enzyme expression levels."
- Figure 6A is difficult to understand and Figure 6A Legend makes it more difficult.
- Odd sentence above Figure 6 (since line numbers are not given in the draft): " Consequently, one can use the SEAMAP to guide the selection of a regulated promoter to dynamically control a pathway's"

Reviewer #2:

The authors have adequately addressed all of my previous concerns.

Reviewer #3:

The author responded to my comments.

2nd Revision - authors' response

27 April 2014

(next page)

Subject: Response to Reviewers

We thank the reviewers for their rapid response to our resubmitted manuscript. Based on the reviewers' comments, we have made additional changes to the manuscript's text and schematics to clarify a few remaining questions. In response to reviewer #1's request, we have also created a new subsection (Box 1) describing our design rules for engineering bacterial operons to achieve proportional control of expression. Box 1 displays a figure showing our design rules in action.

Reviewers #2 and #3 were satisfied with the modifications to our manuscript.

Reviewer #1:

The summary is that the paper has enough strong results to merit publication in MSB. However, there are a number of issues that make the delivery of the main points of the paper difficult to extract and evaluate. While we don't want to force another round of major revision we hope the authors will consider the points below very carefully.

The authors claim that they can use SEAMAP to "guide the selection of a regulated promoter to dynamically control a pathway's." (this sentence is also grammatically incorrect). Where is the data to support this claim?

The sentence has been corrected by adding "activity" to its end. This paragraph now reads, "We then characterized the four pathway variants' productivities with increasing IPTG induction (**Supplementary Table S12**). Though the pathway variants were expressed by the same promoter, their activity responses varied greatly and were highly consistent with model calculations (**Figure 5C**, right). Pathway variants with optimal translation rates had higher productivities, and achieved maximum activity at a lower transcription rate. However, additional increases in transcription lowered their productivities, due to excess enzyme expression levels. The SEAMAP shows how changing the operon's transcription and translation rates can exhibit this non-linear behavior. Consequently, one can use the SEAMAP to guide the selection of a regulated promoter to dynamically control a pathway's activity. Regulated promoters can often serve as sensors for cellular stress, and they may be used to implement feedback control over a pathway's enzyme expression levels to maintain maximal activities. The use of dynamic regulation has been shown to significantly improve a pathway's productivity (Dahl et al, 2013; Zhang et al, 2012)."

As shown in **Figure 5**, the SEAMAP correctly predicts how the pathway's activity will change when altering either the transcription rate of the promoter or the translation rates of the ribosome binding sites. The SEAMAP guides the selection of a promoter by predicting the pathway's activity when the promoter's transcription rate has changed.

Nonetheless, the core of the manuscript is quite interesting and important. The authors design and predict RBS parts performance (translation initiation rate), use them to build synthetic operon and

predict the output of metabolic pathway. Overall, the designability and predictability of synthetic operon are the most important aspects of this work.

The authors provide no explanation on which design specifications went into operon engineering. ... snip ... In the original manuscript, a paragraph in the Discussion discussed design criteria, stated that "Potentially confounding interactions that affect protein expression are minimized by eliminating long single-stranded RNA regions or long RNA duplexes that may reduce mRNA stability, by ensuring that translation elongation is not rate-limiting, and by ensuring that mRNAs are always translated to protect them from RNase activity. By incorporating these design rules into the engineering of bacterial operons, one can achieve proportional control of protein expression by manipulating only RBS sequences. As additional biophysical rules continue to be developed (Espah Borujeni et al), they are incorporated into the forward design process, and can improve the accuracy of predictions on previously designed sequences. Thus, computational design can evolve concomitantly with our understanding of gene expression and the development of new DNA assembly, genome mutagenesis, and genome synthesis techniques to accelerate the engineering of large genetic systems."

In the first version of the manuscript, we had included a paragraph stating additional design criteria to achieve proportional control of expression in bacterial operons in addition to the rational design of RBS libraries. Based on the reviewers' previous comments, the manuscript's text was re-focused on the main conclusions, which did not include these design criteria. However, with reviewer's new comment and at the suggestion of the editor, we have provided the proposed design criteria within the following new subsection, **Box 1**, together with a new figure illustrating these design rules in action.

Box 1: Design Rules for Synthetic Bacterial Operons

Protein expression levels are affected by several factors, including transcription rates, mRNA stabilities, and translation rates. Proportional control over expression is achieved by optimizing RBS libraries to vary translation initiation rates, while carrying out rational sequence design to minimize changes in other factors. Codon usages are optimized to increase their translation elongation rates, while reducing the number of internal start codons with high translation initiation rates (Quan et al, 2011).

Changes in mRNA stability are reduced by shortening unprotected mRNA regions and by removing potential RNase binding sites, including long single-stranded or duplexed RNA regions (Baker & Mackie, 2003; Dasgupta et al, 1998; Folichon et al, 2003; Saito & Richardson, 1981). Intergenic regions are designed to limit the extent of translational coupling within multi-cistronic bacterial operons (Oppenheim & Yanofsky, 1980). Using these rules, the effects of confounding control variables are minimized, thereby enabling designed RBSs to proportionally alter a protein's expression level, regardless of its location within a bacterial operon. Importantly, building and using SEAMAPs employs a

reference genetic system variant; its predictions depend only on proportional changes to protein expression.

The following reviewer's questions are related to the design rules, stated in Box 1.

We would like to know were these biophysical rules applied or not. From the current 'main text', 'supplementary material' and 'methods' section it doesn't appear that these rules have been applied.

The design rules were applied to the synthetic *crtEBI* bacterial operon variants.

If yes- then how did they do it? How did they classify which RNA duplexes reduce mRNA stability and which ones improve the transcript stability? If the authors did use SOME method to "eliminate long single-stranded RNA regions" between intergenic regions, how did they modulate/regulate the low ribosome traffic in the case of weaker RBS1-gene1 design, which will have naked mRNA and may lead to faster degradation of downstream transcript. This may also impact termination of transcription process as both transcription and translation are coupled in E coli.

The lengths of RNA duplexes were counted according to the number of adjacent RNA base pairings as determined by their predicted minimum free energy structure. RNA duplexes longer than 8 bp were classified as potential RNase binding sites. The lengths of single-stranded RNA regions were counted according to the number of adjacent non-paired RNA nucleotides as determined by their predicted minimum free energy structure. ssRNA regions longer than 12 nt were classified as potential RNase binding sites. Only RNA duplexed regions or single-stranded RNA regions unprotected by initiating or elongating ribosomes are counted. These design criteria are based on several studies that have elucidated the sequence specificities of RNase E and RNase III (Baker & Mackie, 2003; Dasgupta et al, 1998; Folichon et al, 2003; Jarrige et al, 2001; Kushner, 2002; Saito & Richardson, 1981).

We really do not understand how the authors "ensured that mRNAs are always translated to protect them from RNase activity" especially, when the strength of RBS1 and RBS2 are weaker than RBS3; and other combinations of such. Do we really know of any proven design rules that can make any mRNA always translated insulated from neighboring context?

The extent of ribosome-dependent protection of mRNA is controlled by minimizing the lengths of untranslated regions to include only the standby site and the ribosome's footprint region, both essential for translation initiation, and to ensure that the rate of translation initiation is sufficiently high so that elongating ribosomes are sufficiently covering translated regions. As shown in **Figures 1**, **Figure 2**, and **Figure 3**, a translation initiation rate between 10 and 100 au was sufficient to protect translated regions from RNase activity, as evidenced by the proportional relationship between predicted translation initiation and protein expression level. Each translated region in the characterized *crtEBI* operon variants have translation initiation rates above 100 au; all these RBSs are sufficiently strong.

We had asked in earlier version of manuscript, a series of questions about how various design specifications were considered in building an operon; how performance of single cistron changed when cloned into an polycistronic operon; how does gene position impact operon or pathway performance and how RBS sequences designed for one gene (and cloned in an operon context) impact the expression of

other genes of operon? How do the authors take into account 'rho-dependent transcription termination activity' when ribosome traffic is minimal for weaker upstream translation events?

It is important to emphasize that our approach to searching, mapping, and optimizing genetic system requires proportional, and not absolute, control over protein expression. Our kinetic model uses a characterized operon variant as its reference state, and its equations are de-dimensionalized into ratios accordingly. The molecular interactions mentioned by the reviewer could affect the absolute amounts of proteins expressed (e.g. mRNA levels being lowest at the end of a transcript, due to RNA polymerase fall off). However, even in the presence of these interactions, a 10-fold higher translation initiation rate will increase the protein's expression level by about 10-fold. Therefore, it was not necessary to account for rho-dependent transcriptional termination to obtain proportional control over protein expression. It was not necessary to account for position-dependent changes in mRNA level, due to RNA polymerase fall off and/or gene order, to obtain proportional control over protein expression.

How does stronger RBS1-gene1 impact expression of downstream weak RBS2-gene2 ? or how does weaker RBS1-gene1 impacts expression of downstream stronger RBS2-gene2 and terminal medium RBS3-gene3?

We agree that accurately predicting expression levels within the context of bacterial operons is an important question. In our recently published study, we measured the effects of newly discovered interactions between the ribosome and long, structured 5' UTRs, incorporating these interactions into an improved biophysical model of translation initiation (Borujeni et al, 2013). Across 136 characterized sequences, the model's predictions correctly explain how changes in 5' UTR sequence, particularly when promoters are replaced, can affect translation initiation rates. In another unpublished study, we have systematically characterized the effects of ribosome-ribosome translational coupling on protein expression levels in multi-cistronic bacterial operons. 165 bacterial operon variants were characterized to formulate a predictive biophysical model that determines how the translation rates of upstream genes could affect the translation rates of downstream genes. This model answers many of the reviewer's questions regarding how translational coupling controls co-dependencies within operons. This model also suggests some common sense rules for minimizing translational coupling, as desired here. These rules include (i) the design of bacterial operons with intergenic distances of at least 20 nt; (ii) avoiding the formation of inhibitory secondary structures that are unfolded by upstream elongating ribosomes; and (iii) removing any extraneous start codons or open reading frames.

How did the authors make sure that "translation elongation is not rate-limiting"?

In the previous version of the manuscript, we presented data showing the effect when translation elongation is rate-limiting; there is a plateau in protein expression at a critically high translation initiation rate. If the same phenomenon were to occur for the *crtEBI* operon variants, one should expect to see a plateau in metabolic activity at a critically high translation initiation rate for *crtE*, *crtB*, or *crtI*. Based on our data, we do not observe a plateau in metabolic activity until translation initiation rates are >100000 au, suggesting that translation elongation rates are at least not rate-limiting up to this very high translation initiation rate. A plateau in metabolic activity would also be observed if precursor biosynthesis was rate-limiting. We tested this hypothesis and found that precursor biosynthesis rates

were the rate-limiting step for an optimally balanced *crtEBI* operon variant with high translation initiation rates at the appropriate ratios (**Figure 6**).

There are many published studies that have attempted, with poor success, to design a predictably operating synthetic operon with genes coding for metabolic pathway enzymes.

Yes, our study shows that systematic variation of translation initiation rate within a well-designed synthetic operon will have predictable effects on the metabolic pathway's activity. Importantly, the pathway's activity is controlled by all the enzymes' expression levels, and in a non-linear fashion. Large changes in multiple enzyme expression levels may be needed to observe a large change in pathway activity. This effect is explained by our model of the pathway's biochemical reactions, in which all reactions are reversible and are modeled using elementary mass action kinetics. Therefore, previous investigations could potentially have failed to observe large changes in pathway activity because enzyme expression levels were individually varied and/or varied over a modest range.

OTHER POINTS:

- *No line numbers and page numbers. So it is challenging to review the manuscript (second time!).* Page numbers have been added.

- *Results are presented in the discussion section*

The paragraph regarding evolutionary dynamics has been moved into the Results section.

- *The authors use the word 'optimized RBS library' and do not really explain what do they mean by 'optimized'?*

The RBS Library Calculator uses an optimization algorithm to design synthetic RBS libraries according to a design objective function. RBS libraries that have been optimized by the RBS Library Calculator have maximized their design objective function. Our terminology and the objective function definition are clearly defined in the Methods section.

- *It seems unlikely that "...these observations suggest that the free energies of in vivo RNA-RNA interactions remain the same regardless of the host organism, including the effects of molecular crowding on binding events." There are many papers showing that host context changes the behavior of synthetic pathways. This conclusion seems contrary to the findings of these studies. Again, there should be a paper dedicated to exploring this topic in detail.*

The term "host context" is very broad as it could mean any host-dependent presence/absence of enzymes, cofactors, or substrates that would affect pathway activity. Here, the free energies of *in vivo* RNA-RNA interactions refers to the amount of available energy released when pairs of RNA molecules form an RNA complex. Our biophysical model performs a set of calculations to determine the Gibbs free energy changes during translation initiation. Our experimental results strongly support the conclusion that these Gibbs free energy changes are the same regardless of the host organism.

- *Figure 2 legend has repeat (B) section.*

Fixed.

- *Figure 4 legend A. where is the 'search' mode (left) and 'zoom' mode (right)?*

Fixed.

- *Throughout manuscript authors have used varied predicted translation rates and do not clarify the decision to do so. For example, under 'Mapping the sequence-expression-activity space of a multi-enzyme pathway' section, ' optimized RBS libraries were designed to vary CrtE, B and I from 445 to 72000 au, 3 to 20,000 au and 97 to 203000 au'. Here, why and how are these lower values chosen?*

Our goal was to maximally alter translation initiation rates across a wide range with the constraint that the maximum translation initiation rate should be very high.

- *Odd sentence above Figure 6 (since line numbers are not given in the draft): "According to the model, the pathway variants' productivities will increase with promoter induction (Figure 5C, left), though variants with optimal translation rates will have higher productivities, and at lower transcription rates"*

This sentence has been rephrased.

- *Did the authors measure any enzyme levels? Sentence above Figure 6 (since line numbers are not given in the draft): "Pathway variants with optimal translation rates had higher productivities, and achieved maximum activity at a lower transcription rate. However, additional increases in transcription lowered their productivities, due to excess enzyme expression levels."*

Protein expression levels were measured to be proportional to the transcription rate of this promoter (**Supplementary Table S12**). Thus, at higher transcription rates, there will be higher protein expression levels. At very high transcription rates, we observed a decrease in pathway productivity.

- *Figure 6A is difficult to understand and Figure 6A Legend makes it more difficult.*

The Figure 6A legend has been modified to include the following, "A lower FCC indicates that the enzyme is less rate-limiting." Please consult (Fell, 1992) for more information.

- *Odd sentence above Figure 6 (since line numbers are not given in the draft): " Consequently, one can use the SEAMAP to guide the selection of a regulated promoter to dynamically control a pathway's"*

Related to the above comment, this sentence was missing the word "activity" at the end. Fixed.

Sincerely,

Howard Salis, Assistant Professor of Biological Engineering and Chemical Engineering
 Pennsylvania State University

References

- Baker KE, Mackie GA (2003) Ectopic RNase E sites promote bypass of 5'-end-dependent mRNA decay in *Escherichia coli*. *Molecular microbiology* **47**: 75-88
- Borujeni AE, Channarasappa AS, Salis HM (2013) Translation rate is controlled by coupled trade-offs between site accessibility, selective RNA unfolding and sliding at upstream standby sites. *Nucleic acids research* **42**
- Dahl RH, Zhang F, Alonso-Gutierrez J, Baidoo E, Batth TS, Redding-Johanson AM, Petzold CJ, Mukhopadhyay A, Lee TS, Adams PD (2013) Engineering dynamic pathway regulation using stress-response promoters. *Nature biotechnology*
- Dasgupta S, Fernandez L, Kameyama L, Inada T, Nakamura Y, Pappas A, Court D (1998) Genetic uncoupling of the dsRNA-binding and RNA cleavage activities of the *Escherichia coli* endoribonuclease RNase III—the effect of dsRNA binding on gene expression. *Molecular microbiology* **28**: 629-640
- Fell DA (1992) Metabolic control analysis: a survey of its theoretical and experimental development. *Biochemical Journal* **286**: 313
- Folichon M, Arluison V, Pellegrini O, Huntzinger E, Régnier P, Hajnsdorf E (2003) The poly (A) binding protein Hfq protects RNA from RNase E and exoribonucleolytic degradation. *Nucleic acids research* **31**: 7302-7310
- Jarrige AC, Mathy N, Portier C (2001) PNPase autocontrols its expression by degrading a double-stranded structure in the pnp mRNA leader. *The EMBO journal* **20**: 6845-6855
- Kushner SR (2002) mRNA decay in *Escherichia coli* comes of age. *Journal of bacteriology* **184**: 4658-4665
- Oppenheim DS, Yanofsky C (1980) Translational coupling during expression of the tryptophan operon of *Escherichia coli*. *Genetics* **95**: 785-795
- Quan J, Saaem I, Tang N, Ma S, Negre N, Gong H, White KP, Tian J (2011) Parallel on-chip gene synthesis and application to optimization of protein expression. *Nature biotechnology* **29**: 449-452
- Saito H, Richardson CC (1981) Processing of mRNA by ribonuclease III regulates expression of gene 1.2 of bacteriophage T7. *Cell* **27**: 533-542
- Zhang F, Carothers JM, Keasling JD (2012) Design of a dynamic sensor-regulator system for production of chemicals and fuels derived from fatty acids. *Nature biotechnology* **30**: 354-359